**METHOD**

# Carnelian uncovers hidden functional patterns across diverse study populations from whole metagenome sequencing reads

Sumaiya Nazeen[1], Yun William Yu[2,3] and Bonnie Berger[1,4*] ⓘ

## Abstract

Microbial populations exhibit functional changes in response to different ambient environments. Although whole metagenome sequencing promises enough raw data to study those changes, existing tools are limited in their ability to directly compare microbial metabolic function across samples and studies. We introduce Carnelian, an end-to-end pipeline for metabolic functional profiling uniquely suited to finding functional trends across diverse datasets. Carnelian is able to find shared metabolic pathways, concordant functional dysbioses, and distinguish Enzyme Commission (EC) terms missed by existing methodologies. We demonstrate Carnelian's effectiveness on type 2 diabetes, Crohn's disease, Parkinson's disease, and industrialized and non-industrialized gut microbiome cohorts.

**Keywords:** Comparative functional metagenomics, Functional profiling, Metagenomic binning, Alignment-free binning, Compositional gapped binning

## Background

Recent advances in next-generation sequencing (NGS) technologies and large-scale national and international efforts [1, 2] have generated unprecedented amounts of microbial genomic data; the NIH's National Center for Biotechnology Information (NCBI), the European Bioinformatics Institute (EBI), and the Joint Genome Institute (JGI) currently host an order of magnitude more shotgun metagenomic data than they did 10 years ago [3]. Many methods have been developed for the initial analyses of this data—assembly, taxonomic binning, and functional profiling of metagenomic reads [3–5] to enable quantitative comparison of microbial communities. Here, we turn our attention to the discovery of trends in microbial metabolic function across diverse populations (different nations or geographical boundaries) with respect to health and disease.

Hundreds of recent studies have implicated the microbiome—which plays a crucial role in the human immune system and metabolism—in complex human diseases. Examples of such diseases include Crohn's disease [6], obesity [7], type 2 diabetes (T2D) [8, 9], colorectal cancer [10], Parkinson's disease (PD) [11], and even autism spectrum disorder (ASD)—which has been found to have an innate immunity component [12, 13]. Many efforts have sought to uncover shared taxonomic dysbiosis (i.e., microbial imbalance) between study populations for a given disease; however, these attempts have generally not found shared taxonomic dysbiosis, probably because the background healthy microbiomes differ significantly in taxonomic composition, to begin with [14–17]. Because different species may fill the same ecological niche, the traditional focus on taxonomy can lose sight of the *functional* relatedness of the microbiomes of two individuals—i.e., commonalities and differences in the functional capabilities of microbial populations [18]. For example, while most strains of lactobacilli exhibit galactosidase activity, that particular functionality can also be partially substituted for by many taxonomically distinct strains of bifidobacteria and bacteriodes [19]. In the large meta-analyses cited above [14–17], there was some attempt to perform functional profiling (in addition to taxonomic profiling), but due to limitations in the study design and methods available, they were unable to find concordant

*Correspondence: bab@mit.edu
[1]Computer Science and Artificial Intelligence Laboratory, MIT, 77 Massachusetts Ave, MA 02139, Cambridge, USA
[4]Department of Mathematics, MIT, 77 Massachusetts Ave, MA 02139 Cambridge, USA
Full list of author information is available at the end of the article

pathways, which one *would* expect from the same disease. Thus, better functional profiling is important to uncovering trends in functional relatedness when comparing study cohorts; this remains an unsolved challenge due to inconsistencies and incompleteness of annotations of microbial genes across reference databases and the lack of comparability of existing relative abundance statistics across samples and studies [3, 20].

To uncover functional trends in microbiomes, an essential first building block is functional profiling of metagenomic reads, the task of assigning reads to known biological function (catalytic action, functional domain categories, genes, etc.) and estimating abundances of those functional terms. Traditional whole metagenome functional annotation approaches assemble reads into large contigs and annotate them using sequence homology, often using existing alignment tools such as BLAST [21], profile Hidden Markov Models (HMMs), or position-specific weight matrices (PWMs). Such methods include RAST [22], Megan4 [23], MEDUSA [24], Tentacle [25], MOCat2 [26], IMG4 [27], and gene catalogue-based methods [2, 28]. Since assembly is a slow, resource-heavy, and lossy process, annotating reads directly via sequence homology or read mapping is used by another class of tools, including MG-RAST [29], HUMAnN [30], ShotMap [31], Fun4Me [32], mi-faser [33], and HUMAnN2 [34]. However, alignment-based read mapping remains time consuming when comparing hundreds of samples from different disease conditions [35, 36]. HUMAnN2 and mi-faser significantly speed up the alignment step by using a fast protein aligner, DIAMOND [37], and thus are able to accurately and quickly capture function from sequences corresponding to known proteins. However, because they are based on alignment, they are challenged in capturing shared features of functionally similar proteins that are not-so-sequence-similar, multi-domain proteins, and remote homologs.

Naturally, predicting function without having experimentally characterized a protein is difficult and runs the risk of false positives. For well-studied populations, there is little need to do so. Many traditional methods for resolving novel protein function depend implicitly on structure, but we currently do not have much structural information available for prokaryotic proteins; thus, we instead approach the problem using sequence correlations. However, when analyzing data from less studied populations—so often the case in metagenomic analysis, a large fraction of reads sequenced do not directly correspond to proteins of known species [18, 38]; thus, sequence-based methods that depend on alignment do not perform as well. We observe this problem when studying the non-industrialized Baka population ("Results" section). Techniques from the field of remote homology detection can be used to explicitly guess at shared functions between an unknown protein and an existing one, but they operate at the level of entire protein sequences, rather than Whole Metagenome Shotgun (WMS) sequencing reads.

Alternately, (gapped) *k*-mer-based taxonomic binning methods have shown great utility compared to read-alignment approaches in assigning reads to taxonomic units [39–42]. Importantly, they can be trained to directly classify WMS reads by function, even when the read itself comes from a protein that is not in existing databases. Using these techniques, we pursue the intuition that we can, for example, predict that reads correspond to a particular enzymatic function (e.g., galactosidase activity) even when the training set does not include the protein from which those reads were taken, but only for distantly related proteins (Additional file 1: Note S1). In this manner, we have the advantage over prior work of being able to predict function without needing explicit alignments.

Importantly, the design of these classification tools allows us to easily construct negative examples during training time to control the false positive rate while still allowing labeling of reads for which alignment is insufficient. Our work thus newly repurposes gapped *k*-mer binning techniques to directly perform efficient and accurate *functional* binning, which performs much better than existing functional profilers based on either alignment or assembly for *analyzing functional relatedness across diverse microbiomes.*

To this end, we introduce Carnelian, a compositional tool for metabolic functional profiling of whole metagenome sequencing reads, and an end-to-end pipeline that is uniquely suited to finding common functional trends across metagenomic datasets from different study populations. The pipeline we present is better suited for "comparative functional metagenomics" for three reasons. First, Carnelian makes use of a gapped *k*-mer classifier [42, 43], which is better able to detect the ECs (Enzyme Commission terms that classify proteins by their enzymatic action) present in non-annotated species, while simultaneously avoiding forced spurious labels through training on a negative set. Second, we build a comprehensive database focused on comparing metabolic functionality, as opposed to using typical protein databases that contain non-prokaryotic and non-metabolic annotations. Third, we present a principled statistical significance analysis for finding shared metabolic pathways using the results of EC detection.

Here, we demonstrate Carnelian's effectiveness through analyses of several real published and unpublished datasets. First, we compare geographically separated study cohorts of type 2 diabetes (T2D) and Crohn's disease (CD). Several of today's state-of-the-art functional annotation tools, including mi-faser, HUMANn2 (translated search), and Kraken2 (protein search), were unable to find concordant functional dysbioses between healthy and

diseased microbiomes, which one would expect given that the same disease should have similar effects on different study populations. Importantly, Carnelian is able to find those expected concordant functional dysbioses. Next, we find that Carnelian-identified EC terms can classify patients vs. controls consistently, with higher accuracy than existing tools across T2D, CD, and Parkinson's disease (PD); this finding suggests that the additional ECs identified by Carnelian are not spurious. Next, using a combination of published and unpublished datasets, we further demonstrate Carnelian's effectiveness on geographically and dietarily diverse healthy microbiomes of industrialized individuals from the USA (Boston) [44] and non-industrialized communities from Cameroon (Baka ethnicity), Ethiopia (Gimbichu region) [38], and Madagascar (Betsimisaraka and Tsimihety ethnicities) [38]. Unlike existing methods, Carnelian was able to uncover the expected pathway-level similarities in core metabolic function between healthy individuals from each of those communities. Lastly, on a Parkinson's disease case-control metagenomic dataset, we show that Carnelian uniquely finds several hallmarks of Parkinson's disease in the patient microbiomes. As new data is collected from all over the world (e.g., our Cameroon data), we expect Carnelian to be an essential tool in analyzing functional similarities and differences across diverse populations.

## Results

We present Carnelian, a novel gapped *k*-mer-based functional profiler, and an end-to-end pipeline for comparative functional metagenomic studies using WMS reads from diverse study populations. Our pipeline enables the comparison of functional summaries of WMS data by designing more consistently annotated reference databases of microbial proteins, building a functional annotation tool better suited for assigning functions to reads that are not readily alignable to known proteins, and generating comparable abundance statistics across samples and studies (Fig. 1).

WMS data comes from a mixture of organisms, and can encode 100x more unique genes than those present in just the human genome [28]. Only a fraction of these genes have known functional annotations in existing databases. Even of those genes with annotations, many of the annotations are computationally predicted and therefore less reliable. We are also primarily interested in microbial functions that can influence host health, such as production of metabolites, extracellular enzymes, or immunostimulatory surface structures [45]. Thus, we constructed our gold standard reference database with curated prokaryotic proteins that have verified unique and complete EC labels which provide a direct mapping to KEGG metabolic pathways for our later analyses. Our curated database consists of 7884 prokaryotic

proteins with 2010 unique EC labels and is provided on our website.

Another important characteristic of metagenomic data is that the reads sequenced often come from non-annotated species; without a known reference, taxonomic read classifiers are limited in their annotation ability. Luckily, related proteins that share a function also share compositional (gapped *k*-mer) features in their amino acid sequence, even across species. Leveraging this intuition, the Carnelian pipeline uses probabilistic ORF detection to enable application of a compositional gapped classifier ensemble on the full amino acid sequence; this classifier ensemble is better able to bin proteins present in non-annotated species. More precisely, Carnelian first detects all possible ORFs from the input reads using FragGeneScan [46], which probabilistically detects the coding part(s) of the reads. Then, Carnelian encodes the ORFs into a low-dimensional compact feature space using Opal-Gallager hashes [42, 43]. Once so encoded, these ORFs are annotated by Carnelian's classifier ensemble, a set of one-against-all support vector machines. The classifier ensemble is trained with functionally annotated gold standard proteins represented in the same compact feature space, and with negative samples based off of randomly shuffling in human sequences generated via HMMER [47]. The training is performed in an online fashion (i.e., only one input sequence is loaded in memory at a time), making incremental training of Carnelian's classifier ensemble easy when new annotations become available.

Relative abundance statistics output from standard functional profiling tools are not directly comparable across samples and studies; to address this problem, Carnelian borrows from transcriptomic normalization practices. From input WMS reads, Carnelian constructs a functional vector containing effective read counts per EC label (i.e., read counts normalized against effective protein length per EC label and a per million scaling factor that takes into account the effect of the lengths of proteins with other EC labels on the relative abundance of a particular EC label) ("Methods" section). This normalization step is similar to the "transcripts per million" (TPM) counts used for quantifying transcript abundances from RNA-seq data [48]. The sum of Carnelian's effective read counts thus remains constant across all samples, unlike the raw read counts and reads per kilobase (RPK) measures used by existing functional annotation tools. This normalization makes sample profiles directly comparable to each other across experiments performed with different sequencing depths (see Additional file 1: Note S2 for a demonstration of the lack of comparability when no normalization or RPK normalization is done.) These EC profiles are used to quantify KEGG metabolic pathways ("Methods" section) for comparative analysis of different study populations.

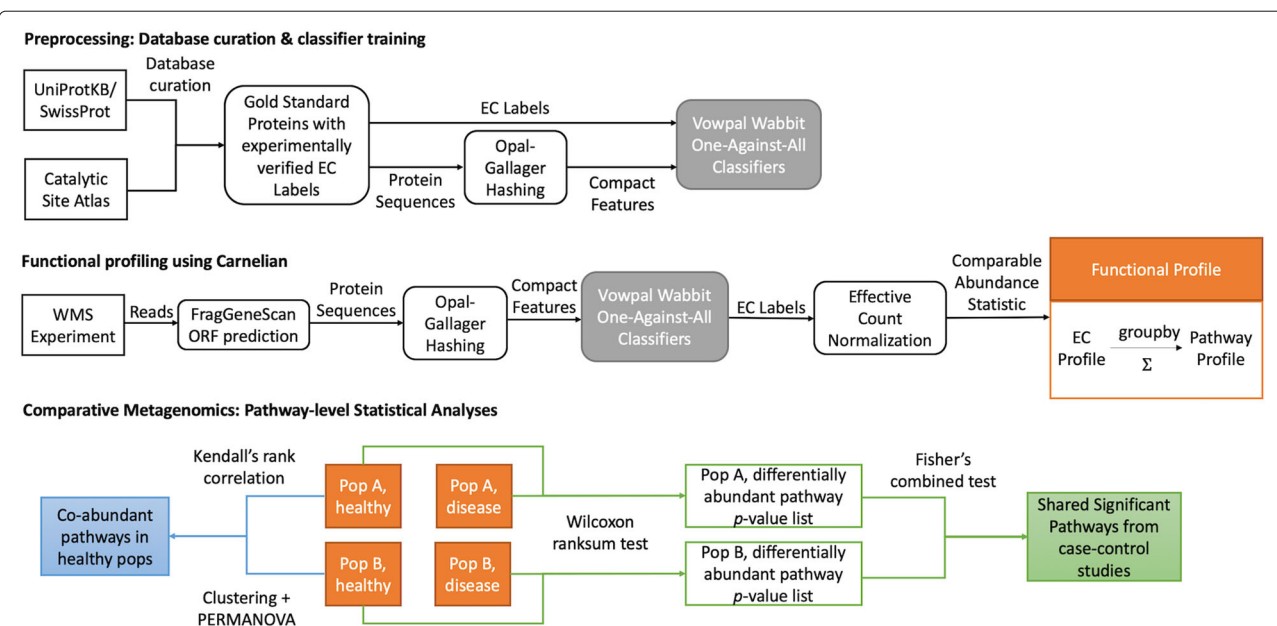

**Fig. 1** Comparative functional metagenomics with Carnelian. *Preprocessing.* We build a gold standard reference database by combining reviewed prokaryotic proteins with complete Enzyme Commission (EC) labels and evidence of existence from UniProtKB/Swiss-Prot with curated prokaryotic catalytic residues with complete EC labels from the Catalytic Site Atlas. Carnelian first represents gold standard proteins in a compact feature space using low-density, even-coverage locality-sensitive Opal-Gallager hashing. Then, it trains a set of one-against-all (OAA) classifiers (implemented using the Vowpal Wabbit framework) using the compact feature representation of those proteins as well as negative samples based off of random shuffled sequences generated by HMMER. *Functional profiling.* To functionally profile reads from a whole metagenomic sequencing (WMS) experiment, Carnelian first performs probabilistic ORF prediction using FragGeneScan. Next, the ORFs are represented in a compact feature space using the same Opal-Gallager hashing technique. The trained OAA classifier ensemble is then used to classify the ORFs into appropriate EC bins. Abundance estimates of ECs are computed from the raw ORF counts in the EC bins by normalizing against effective protein length per EC bin and a per million scaling factor. Pathway profiles (Orange) are computed by grouping the ECs into metabolic pathways and summing the abundance estimates. *Comparative metagenomics.* We start from pathway profiles (Orange) of different populations and conditions. (Blue) Functional relatedness of healthy microbiomes across different populations is assessed by co-abundance pathway analysis. Pathway co-abundance estimates are quantified by Kendall's rank correlation. Co-abundance clusters are determined by Ward-Linkage hierarchical clustering, and the PERMANOVA test is used to determine if the centroids of those clusters differ between populations A and B. (Green) Functional trends analysis across different case-control cohorts of a disease is performed using differential abundance analysis by Wilcoxon rank-sum test and shared significance analysis by Fisher's combined probability test

Carnelian is robust to sequencing technology biases (Additional file 1: Note S3) and is equally applicable to non-human metagenomic datasets where it can find meaningful biological patterns (Additional file 1: Note S4). Carnelian achieves higher sensitivity and F1-score than current state-of-the-art alignment-based tools: mi-faser [33] and HUMAnN2 [34] (translated search) as well as a fast alignment-free *k*-mer-based tool Kraken2 [39] (protein search)—all run on the same in-house benchmarks (Additional file 1: Note S5). On a synthetic human gut metagenomic dataset of 5 million reads (150 bp, single-ended), Carnelian requires ∼ 16 min using 16 CPU cores—this is roughly 2× faster than mi-faser (∼ 29 min) and similar to HUMAnN2's translated search (∼ 18 min) on the same number of CPUs on the same machine (a 40-core machine with 320-GB RAM, each core was Intel Xeon CPU E5-2695 v2 @ 2.40 GHz). Although Kraken2's protein search is the fastest among all four methods, its performance is significantly worse than

the other methods in terms of sensitivity and F1-score (Additional file 1: Note S5). Unless otherwise stated, all four methods were run with Carnelian's curated reference database for all the experiments to ensure an unbiased comparison.

## Carnelian reveals novel and shared functional dysbiosis across disease studies

Comparing healthy and diseased microbiomes is key to understanding their effect on host biology, enabling clinical diagnoses and informed therapeutics [49, 50]. While taxonomic dysbiosis (i.e., alteration of species-level composition of the microbiome) in patient population is often geography-specific and not generalizable [9, 16, 51], we instead looked at functional dysbiosis. As expected, functional dysbiosis is indeed more generalizable in the type 2 diabetes and Crohn's disease datasets we studied, but only when we used Carnelian as opposed to other methods for the analysis.

We quantified the metabolic functional capacity of the gut microbiomes of patients and controls in two large-scale T2D datasets [8, 9] and two CD datasets [1, 52] at enzyme and pathway levels. Our results revealed concordant functional dysbioses between geographically separated disease cohorts—13 common metabolic pathways between Chinese and European T2D patient microbiomes and 8 common pathways between US and Swedish CD patient microbiomes (Table 1).

For the T2D cohorts, we generated the EC profiles of preprocessed fecal samples from Chinese and European individuals using Carnelian and determined the differentially abundant ECs between patients and controls using a cutoff of Wilcoxon rank-sum test $p$ value <0.05 after Benjamini-Hochberg (BH) correction and absolute log fold change >0.33. In both Chinese and European cohorts, Carnelian reported reduced levels of several glycosyltransferases (e.g. 2.4.1.1, 2.4.1.7, 2.4.1.15) and abundance of several carbon-oxygen lyases (e.g. 4.2.1.120, 4.2.1.20, 4.2.1.42) in the T2D gut (Additional file 2: Tables S1 and S2). At the pathway level, it found 30 significantly altered metabolic pathways in the Chinese T2D patients (BH-corrected Wilcoxon rank-sum test $p$ value <0.05 and absolute log fold change >0.11) and 36 pathways altered between European T2D patients and individuals with normal glucose tolerance (NGT) (Additional file 2:

**Table 1** Shared functional dysbiosis between two type 2 diabetes (T2D) cohorts and two Crohn's disease (CD) cohorts

| ID | Pathway | Carnelian | mi-faser | HUMAnN2 | Kraken2 | Fisher's $p$ (Carnelian) |
|---|---|---|---|---|---|---|
| (a) Common pathways between Chinese and European T2D cohorts | | | | | | |
| 00030 | Pentose phosphate pathway | SB | NB | NB | NB | 6.59E−03 |
| 00040 | Pentose and gluconate interconversions | SB | NB | NB | NB | 9.88E−03 |
| 00051 | Fructose and mannose metabolism | SB | SE | NB | NB | 4.94E−04 |
| 00052 | Galactose metabolism | SB | NB | NB | NB | 4.71E−03 |
| 00061 | Fatty acid biosynthesis | SB | SC | NB | SC | 6.56E−03 |
| 00190 | Oxidative phosphorylation | SB | SE | SC | SE | 4.97E−04 |
| 00250 | Alanine, aspartate, and glutamate metabolism | SB | NB | NB | NB | 1.48E−04 |
| 00290 | Valine, leucine, and isoleucine biosynthesis | SB | SE | NB | NB | 1.68E−05 |
| 00590 | Arachidonic acid metabolism | SB | NB | NB | NB | 2.11E−03 |
| 00600 | Sphingolipid metabolism | SB | SE | NB | SC | 8.86E−05 |
| 00730 | Thiamine metabolism | SB | NB | NB | NB | 2.62E−03 |
| 00983 | Drug metabolism—other enzymes | SB | NB | NB | NB | 2.62E−03 |
| 00195 | Photosynthesis | SB | SB | SC | SB | 2.74E−03 |
| 00254 | Aflatoxin biosynthesis | SC | SC | NB | SB | 1.03E−02 |
| (b) Common pathways between US and Swedish CD cohorts | | | | | | |
| 00500 | Starch and sucrose metabolism | SB | NB | SS | SS | 4.91E−06 |
| 00620 | Pyruvate metabolism | SB | NB | NB | SS | 4.05E−04 |
| 00640 | Propanoate metabolism | SB | NB | NB | NB | 9.04E−03 |
| 00290 | Valine, leucine, and isoleucine biosynthesis | SB | SS | NB | SS | 5.03E−03 |
| 00450 | Selenocompound metabolism | SB | NB | NB | NB | 8.95E−03 |
| 00460 | Cyanoamino acid metabolism | SB | NB | SS | SS | 8.33E−05 |
| 00513 | Various types of N-glycan biosynthesis | SB | NB | NB | NB | 5.79E−03 |
| 00710 | Carbon fixation in photosynthetic organisms | SB | NB | NB | SS | 1.09E−05 |
| 00410 | Beta-alanine metabolism | NB | SS | NB | SB | 5.79E−01 |

(a) Common pathways between Chinese and European T2D cohorts which have significantly altered read abundances. We found 13 shared pathways of which 12 are highly relevant to T2D; these pathways are significant in individual cohorts (BH-corrected Wilcoxon rank-sum test $p$ value < 0.05) as well as in Fisher's combined test at $p$ value < 0.05 cutoff. On the other hand, mi-faser finds only the photosynthesis pathway and Kraken2 finds the photosynthesis and aflatoxin biosynthesis pathways to be commonly disrupted between both the cohorts; with HUMAnN2-profiles, no overlap at the pathway level was found (Additional file 2: Tables S11–S16). (b) Common pathways between the US and Swedish CD cohorts which have significantly altered read abundances. We identify shared dysbiosis in 8 pathways between the two study cohorts; these pathways are significant in individual cohorts as well as in Fisher's combined test at $p$ value < 0.05 cutoff. On the other hand, only Kraken2 finds the beta-alanine metabolism pathway to be commonly disrupted between both the cohorts; with mi-faser- and HUMAnN2-profiles, no overlap at the pathway level was found (Additional file 3: Tables S23, S24, S27, S28, S31, and S32). SB significant in both the studies, NB detected but not significant in both the studies, SC significant in the Chinese cohort only, SE significant in European cohort only, SU significant in the US cohort only, SS significant in the Swedish cohort only

Tables S3 and S4). Notably, 13 of these pathways are significantly shared between both patient cohorts (Fisher's combined $p$ value <0.05) and highly relevant to T2D ((a) in Table 1). For example, we observed significant depletion of reads in several carbohydrate metabolism pathways, such as the pentose phosphate pathway, pentose and glucuronate interconversions, fructose and mannose metabolism, galactose metabolism in patient guts compared to controls in both cohorts (Additional file 2: Tables S3 and S4). Across these two cohorts, we also observed a higher rate of oxidative phosphorylation in the patient gut—a finding that is in agreement with the original studies [8, 9]. Additionally, in each of the patient cohorts, we found significantly lower read abundances in several vitamin-B metabolism pathways (e.g. thiamine metabolism) compared to the healthy gut. Notably, EC- and pathway-level results from mi-faser, HUMAnN2, and Kraken2 were unable to uncover shared pathways of relevance between the two cohorts (Additional file 2: Tables S5-S16).

Carnelian-generated EC profiles of the Crohn's disease cohorts revealed a shift in the metabolic functionality of the patient gut microbiome compared to the control gut microbiome as indicated by lower read abundances in several essential enzymes and pathways. The most significantly variable ECs between patients and controls (Wilcoxon rank-sum test $p$ value <0.05 after BH correction and absolute log fold change >0.58) in both the US and Swedish cohorts include several hexosyltransferases (2.4.1.-), oxidoreductases acting on aldehyde group (1.2.7.-), glycosidases (3.2.1.-), and hydrolyases (4.2.1.-), which are key players in different carbohydrate metabolism pathways (Additional file 3: Tables S17 and S18). Many of these enzymes were not found by other methods. We also observed a decrease in the relative abundance of several enzymes, including aminobutyraldehyde dehydrogenase (1.2.1.19), acetylornithinase (3.5.1.16), lysine decarboxylase (4.1.1.18), and 5-carboxymethyl-2-hydroxymuconic acid isomerase (5.3.3.10). These enzymes play crucial roles in the metabolism of several essential amino acids including arginine, proline, lysine, tyrosine, etc. Thus, this finding might indicate a lower rate of microbial absorption of such amino acids from diet. Several enzymes involved in vitamin B metabolism such as pyridoxine phosphatase (3.1.3.74), dihydroxy-acid dehydratase (4.2.1.9), phosphomethylpyrimidine synthase (4.1.99.17), etc. were also found to be depleted in the CD gut; of the methods we compared against, only Carnelian was able to uncover these findings (Additional file 3: Tables S17 and S18).

At the pathway-level, we found 25 significantly altered metabolic pathways in the guts of CD patients from the US (BH-corrected Wilcoxon rank-sum test $p$ value <0.05 and absolute log fold change >0.11) and 35 pathways

altered between Swedish CD patients and healthy individuals (Additional file 3: Tables S19 and S20). Notably, eight of these pathways are significantly shared between both patient cohorts (Fisher's combined $p$ value <0.05) and seven of them are highly relevant to Crohn's disease ((b) in Table 1). For example, we observed significant depletion of reads in three carbohydrate metabolism pathways, namely, starch and sucrose metabolism, pyruvate metabolism, and propanoate metabolism in patient guts compared to the controls in both the cohorts (Additional file 3: Tables S19 and S20). In both datasets, we also observed lower abundance of reads in valine, leucine and isoleucine (essential amino acids) biosynthesis and cyanoamino acid metabolism pathways in CD patients. We further observed a lower abundance of reads in the selenocompound metabolism and various N-glycan biosynthesis pathways in the CD guts compared to the normal individuals in both cohorts. Although non-specific to CD, the reduced read abundance in carbon fixation pathway might be indicative of the imbalance of energy homeostasis in the patient gut. Importantly, mi-faser and HUMAnN2 found no shared pathways of relevance between the two cohorts and Kraken2 found shared dysbiosis in only the beta-alanine metabolism pathway. EC- and pathway-level results from mi-faser, HUMAnN2, and Kraken2 can be found in Additional file 3: Tables S21-S32.

## Carnelian enables accurate patient vs control classification using functional metagenomic markers

Patients and controls in case-control cohorts of type 2 diabetes (T2D), Crohn's disease (CD), and Parkinson's disease (PD) can be classified with much higher accuracy using the differentially abundant enzyme markers (EC terms) identified by Carnelian, implying that Carnelian's additional labeling of unalignable reads is meaningful. It also implies that the differentially abundant ECs we detect serve as more useful features for the random forest classifier in discriminating patients from controls. To test the power of significantly variable EC terms in discriminating patients from controls in the disease datasets, we performed $N$-fold cross-validation experiments (T2D: tenfold, CD: fivefold, and PD: fivefold). In each trial, ECs exhibiting significant differences in read abundance between patients and controls in the training partition (Wilcoxon rank-sum test $p$ value <0.05) were selected as features input to a set of random forest classifiers and accuracy was measured on the test partition.

In the Chinese T2D cohort, with Carnelian-identified differentially abundant ECs, we achieved an average area under the curve (AUC) of 0.75 (95% CI 0.69–0.82), whereas using the ECs identified by mi-faser, HUMAnN2, and Kraken2, average AUCs of 0.69, 0.63, and 0.63

were achieved, respectively (Fig. 2a). In discriminating European T2D patients from NGT individuals, we achieved an average AUC of 0.72 (95% CI 0.61–0.82) with Carnelian-identified ECs which is significantly higher than the other three methods (Fig. 2b).

In the CD cohort from the USA, we achieved an average AUC of 0.73 (95% CI 0.56–0.89) with Carnelian-

identified differentially abundant ECs, whereas using the differentially abundant ECs identified by mi-faser, HUMAnN2, and Kraken2, average AUCs of 0.61, 0.54, and 0.55 were achieved, respectively (Fig. 2c). In discriminating Swedish CD patients from the healthy controls, we achieved an average AUC of 0.95 (95% CI 0.89–1.00) with Carnelian-identified variable ECs which is

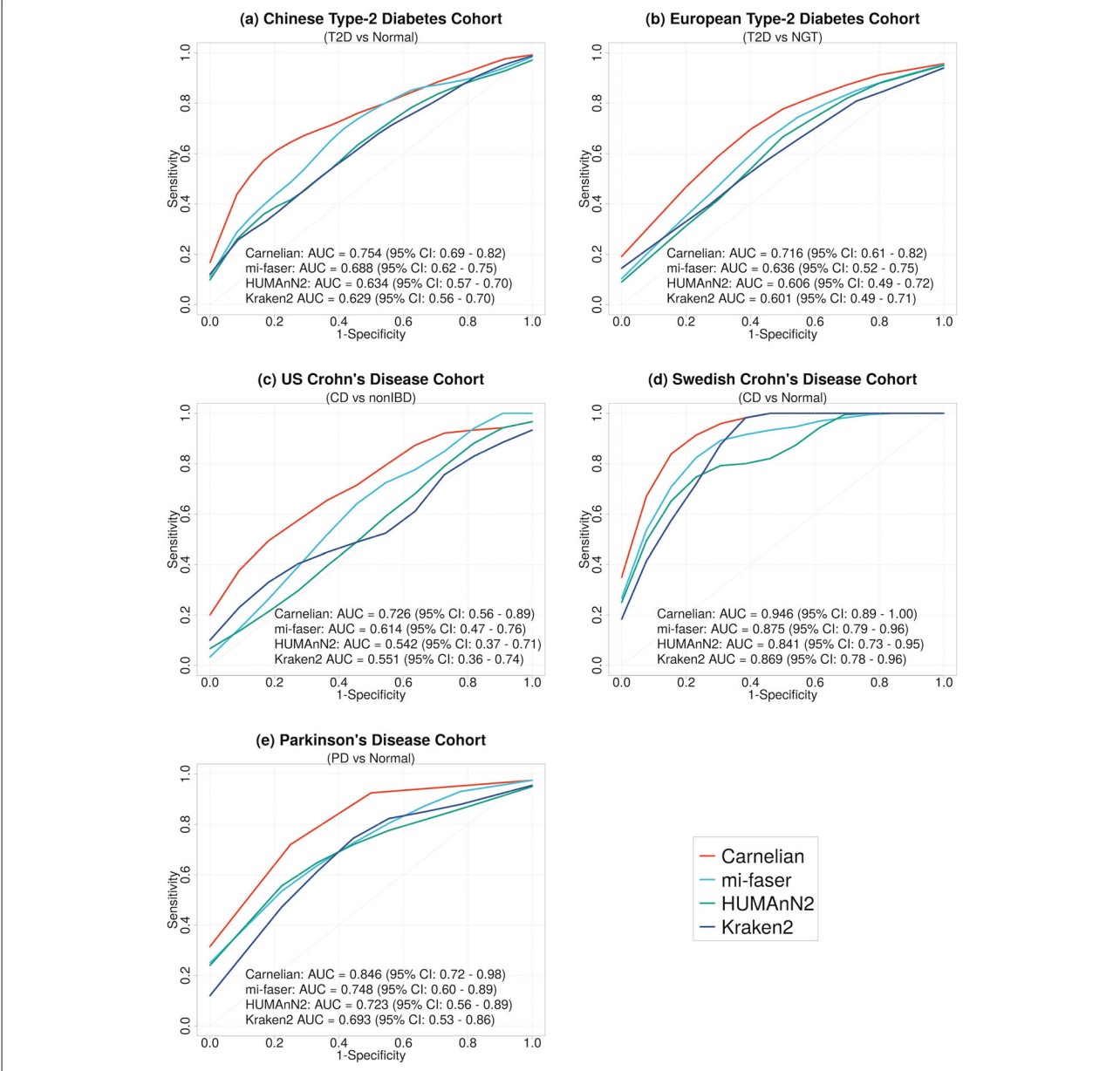

**Fig. 2** Classification of patients vs controls using Enzyme Commission (EC) markers (*N*-fold cross-validation experiments). **a** T2D vs controls in the T2D-Qin dataset (Chinese cohort). **b** T2D vs normal glucose tolerance (NGT) individuals in the T2D-Karlsson dataset (European cohort). **c** CD patients vs controls in the CD-HMP dataset (individuals from the US). **d** CD patients vs healthy individuals in the CD-Swedish dataset (Swedish twin studies). **e** PD vs controls in the PD-Bedarf dataset. In each trial, one of the *N* subsets was selected as the test set and the rest *N* − 1 subsets were used as the training set. Differentially abundant ECs were selected from the training set as features input to a set of random forest classifiers. Performance of classification was measured on the test set. Carnelian-identified EC terms achieve a larger average area under the curve (AUC) in all the cases compared to those identified by other methods

significantly higher than the other three methods (Fig. 2d). In the PD cohort, Carnelian-identified markers achieved an AUC of 0.85 (95% CI 0.72 to 0.98) in discriminating between patients and healthy controls, whereas the differentially abundant EC terms found by other methods did not achieve more than 0.75 average AUC (Fig. 2e).

Note that while it is common practice in the metagenomic literature to select classification features from the entire dataset, even when running cross-validation experiments [8, 9], in all our cross-validation analyses, we instead followed standard machine learning best practices and avoid information leakage in feature selection by choosing EC labels only from the training sets. For completeness, we also performed the classification experiments choosing EC labels from the entire dataset. Using this experimental design, Carnelian-generated ECs again achieved higher accuracy compared to the other three methods in all the study cohorts (Additional file 4: Figure S1).

To test for generalizability of Carnelian-identified ECs in the CD and T2D cohorts, we combined the EC markers identified in the geographically separated cohorts and redid the classification of patients vs controls. For CD, using the unified ECs identified by Carnelian as features, we could achieve $\sim$ 0.94 AUC on average, whereas the combined ECs identified by other tools achieved average AUCs between 0.79 and 0.88 (Additional file 4: Figure S2). For T2D, with the unified ECs from Carnelian as features, we were able to classify the functional profiles of T2D patients and controls with an average AUC of $\sim$ 0.80, whereas using the combined ECs identified by other methods in both cohorts as features, the average AUCs remained between 0.73 and 0.76 (Additional file 4: Figure S3). The lists of combined EC markers for T2D and CD identified by Carnelian are provided in Additional file 5: Tables S33 and S34.

### Carnelian uncovers functional relatedness of diverse industrialized and non-industrialized communities

In addition to finding trends in functional changes across disease cohorts, Carnelian enables us to compare the functional potential of healthy human gut microbiomes from industrialized and non-industrialized communities. We analyzed the fecal microbiomes of 84 individuals from Boston with an urban lifestyle (industrialized society; dataset from the Alm lab [44]), 35 huntergatherer Baka individuals from Cameroon (unpublished dataset from the Alm lab), 50 non-industrialized individuals from Gimbichu, Ethiopia [38], and 112 individuals of Betsimisaraka and Tsimihety ethnicities from Madagascar [38]. The expectation is that healthy individuals across populations ought to share similar core metabolic pathways [18, 53]. Carnelian's analyses met this expectation, finding pathway-level similarity in core metabolic functionality of both the industrialized and non-industrialized communities.

Using our curated EC database, Carnelian detects more ECs compared to other methods (Additional file 5: Table S35) and finds slightly higher diversity of identified ECs in the non-industrialized communities compared to the industrialized community indicated by the Shannon-Wiener diversity index (Additional file 4: Figure S4(a)). At the pathway level, the diversity of identified functionality in both communities is comparable, as hoped (Additional file 4: Figure S4(b)). At both levels, Carnelian captures significantly more diversity than the other three methods (Additional file 4: Figure S4). Importantly, the fecal microbiomes of Baka individuals from Cameroon could not be characterized well even running the full HUMAnN2 pipeline using its default databases. Despite incorporating taxonomic information, out-of-the-box HUMAnN2 could map on average $\sim$ 10% of the reads and detect less than 30 known species and 996 ECs per sample (Shannon diversity index for ECs 5.58) (see Additional file 1: Note S6).

Principal component analysis of the EC profiles generated by Carnelian shows a marked separation by population (Additional file 4: Figure S5(a)). Mean EC profiles of industrialized and non-industrialized microbiomes show a moderate degree of correlation (Kendall's $\tau = 0.75$). Much of this separation washes away at the pathway level (Additional file 4: Figure S5(b)); mean pathway profiles of industrialized and non-industrialized microbiomes show a high degree of correlation (Kendall's $\tau = 0.93$). This finding suggests a high degree of pathway-level functional similarity between industrialized and non-industrialized healthy microbiomes—which was not observed by earlier studies.

In order to identify the ECs that characterize the separation for industrialized and non-industrialized population, we looked at the weights of the ECs in the first nine principal components, which together explain 80% of the variability among individuals (Additional file 5: Table S36). The majority of ECs with high weights were involved in the carbohydrate, amino acid, nucleotide, and energy metabolism pathways. Using the highly weighted ECs, we performed Ward-linkage hierarchical clustering based on Pearson correlation coefficients of the EC profiles of the industrialized and non-industrialized individuals; we observed a clear separation of the two groups (Fig. 3a).

We also identified the significantly variable ECs between the two groups using a cutoff of BH-corrected Wilcoxon rank-sum test $p$ value $< 0.05$ and absolute log fold change $> 1$ (Additional file 5: Table S37). The differentially abundant ECs identified by Carnelian recapitulate the findings of earlier studies; those ECs match the microbial enzymatic functions related to differences in diet

**(a) Separation of Industrialized vs non-industrialized communities at the EC level**

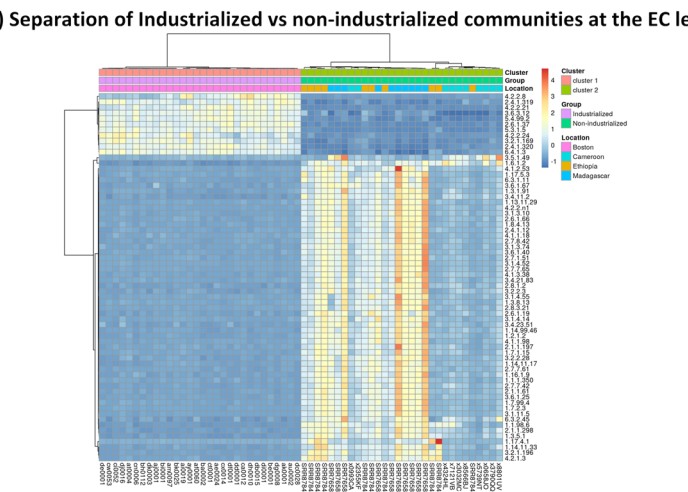

**(b) Co-abundance association across core metabolic pathways:**
**Pathway clusters not significantly different between industrialized and non-industrialized communities**

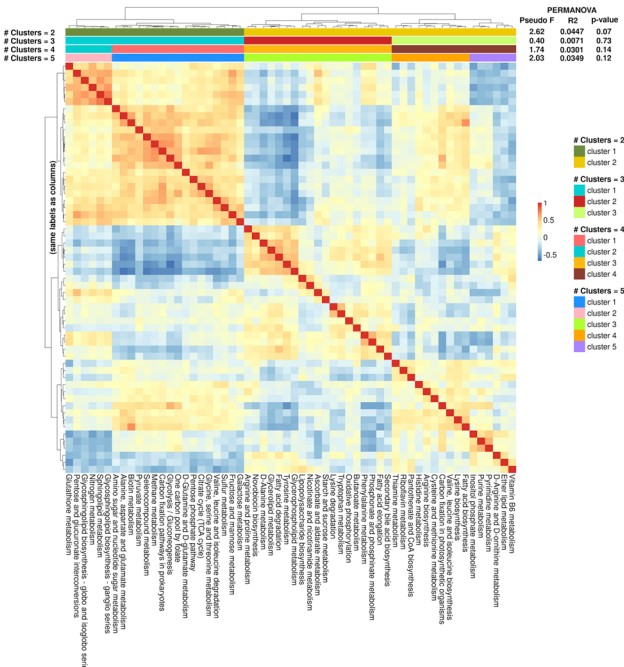

**Fig. 3** Functional diversity and relatedness between industrialized and non-industrialized communities. **a** Heatmap showing the *z*-scores of read abundances of the ECs with high weights in the top principal components. Standard Ward-linkage hierarchical clustering of the EC profiles of industrialized and non-industrialized microbiomes was performed using Pearson correlation. The two top-level clusters found by hierarchical clustering perfectly capture the separation of non-industrialized and industrialized microbiomes. For display purposes, we show only individuals with read abundances falling outside one standard deviation of the mean in at least nine of the highly variable ECs. See Additional file 4: Figure S6 for the corresponding heatmap and clustering on all individuals. **b** Heatmap showing co-abundance association across core metabolic pathways. Co-abundance associations between pathways wee calculated as the pairwise Kendall rank correlations between the pathway abundance profiles (obtained using Carnelian-generated EC profiles) of microbiomes from both communities considered together. Ward-linkage hierarchical clustering was used to partition the pathways using Euclidean distance, generating either 2, 3, 4, or 5 clusters. Although hierarchical clustering can be used to identify clusters of co-abundance pathways between the non-industrialized vs industrialized communities, the clusters were not significantly different from each other with respect to the industrialized/non-industrialized label (PERMANOVA test *p* values > 0.05). Thus, in contrast to the top-level EC label clustering from **a**, the partitions are not simply recapitulating the industrialized/non-industrialized labels

and lifestyle [38, 54, 55]. For example, fecal microbiota from the non-industrialized communities showed over-representation of several enzymes (exclusively identified by Carnelian) involved in the metabolism of fructose, mannose, starch, and sucrose. Examples include mannosyl-3-phosphoglycerate synthase (2.4.1.217), sucrose phosphorylase (2.4.1.7), phosphoglycerate mutase (5.4.2.11), and phosphate propanoyltransferase (2.3.1.222). On the other hand, fecal microbiota of industrialized individuals showed overrepresentation of simple sugar metabolizing enzymes such as ornithine amino-transferase (2.6.1.13), lysine 2,3-aminomutase (5.4.3.2), glycogenase (3.2.1.1), NADP-glucose-6-phosphate dehy-drogenase (1.1.1.49), and phosphohexokinase (2.7.1.11). Urease enzyme (3.5.1.5), which potentially plays a role in synthesizing essential and non-essential amino acids by releasing ammonia as well as a number of amino acid metabolizing enzymes—including ornithine car-bamoyltransferase (2.1.3.3; metabolizes arginine), lysine decarboxylase (4.1.1.18; metabolizes lysine), and lysine racemase (5.1.1.5; metabolizes lysine)—showed higher read abundance in non-industrialized communities (not found by other methods). In addition, Carnelian exclusively found read enrichment in phospholipase D (3.1.4.4; involved in lipid metabolism) and phospho-adenylate 3′-nucleotidase (3.1.3.7; involved in sulfur metabolism), and depletion of phenylacetyl-CoA lig-ase (6.2.1.30; involved in phenylalanine metabolism), pyrrolysyl-tRNA synthetase (6.1.1.26; involved in aminoacyl-tRNA synthesis), and potassium-importing ATPase (3.6.3.12; involved in microbial potassium import) in the non-industrialized communities compared to the industrialized one.

We then explored the co-abundance associations between the core metabolic pathways involved in carbo-hydrate, protein, lipid, energy, and vitamin, and co-factor metabolism. Although hierarchical clustering can be used to identify clusters of co-abundance pathways between the non-industrialized vs industrialized communities, the clusters were not significantly different from each other with respect to the industrialized/non-industrialized label (PERMANOVA test $p$ values $> 0.05$, pseudo-$F$ values close to 1, and small $R^2$ values) (Fig. 3b). This result confirms the existence of pathway-level similarity in the core metabolic functionality (carbohydrate, amino acid, lipid, energy, vitamin, and co-factor metabolism) between the healthy gut microbiomes of non-industrialized and industrialized population.

### Carnelian uncovers novel functional dysbiosis in Parkinson's patient microbiomes

Not only does Carnelian find consistent functional pat-terns in healthy and disease microbiomes across dif-ferent geographies, but it also helps us uncover novel

biology when applied to metagenomic data from a dis-ease with poorly understood links to the gut micro-biome. For example, although two thirds of the patients with Parkinson's disease (a neurodegenerative disease of complex etiology) suffer from gastrointestinal (GI) abnor-malities [56], it is not well understood how the gut microbiome is associated with the disease process. We applied Carnelian on whole metagenome sequencing data from the gut microbiomes of early-stage L-DOPA- naive Parkinson's disease (PD) patients and controls [57] to investigate the differences between the functional capacity of healthy and Parkinson's gut.

Our results reveal a hitherto unobserved functional shift in the gut microbiome of early-stage Parkinson's disease patients from the microbiome of healthy controls through performing differential abundance analyses of ECs and pathways. At the EC level, Carnelian exclusively identi-fies significant variation in read abundance (Benjamini-Hochberg (BH)-corrected Wilcoxon rank-sum test $p$ value $< 0.05$ and absolute log fold change $> 0.58$) in ribonucleoside-diphosphate reductase (1.17.4.1; impli-cated in glutathione metabolism), alpha-galactosidase (3.2.1.22; implicated in lipid metabolism), kynureninase (3.7.1.3; implicated in tryptophan metabolism), etc. (ECs identified by all four methods are provided in Additional file 6: Tables S38–S41). At the pathway level, we found the PD gut to have lower read abundance in several car-bohydrate metabolism pathways (BH-corrected Wilcoxon rank-sum test $p$ value $< 0.05$ and absolute log fold change $> 0.11$) (Additional file 6: Table S42). Differential read abundances in different carbohydrate metabolism pathways were also found by HUMAnN2, mi-faser, and Kraken2 (Additional file 6: Tables S43–S45). Carnelian also identified significantly lower read abundances (BH-corrected Wilcoxon rank-sum test $p$ value $< 0.05$ and absolute log fold change $> 0.11$) in phenylalanine, tyrosine, and tryptophan biosynthesis (missed by both mi-faser and HUMAnN2); alanine, aspartate, and gluta-mate metabolism (missed by HUMAnN2); sphingolipid metabolism (missed by HUMAnN2 and Kraken2); gly-cosphingolipid biosynthesis; and D-alanine metabolism, notably missed by the other three methods (Additional file 6: Tables S42–S45). Note that the original study— which employed an assembly-based functional annota-tion approach using gene catalogs—found differences in gene abundances in only the D-glucuronate and tryp-tophan metabolism pathways [57]. We also analyzed the same dataset with an out-of-the-box HUMAnN2 pipeline with its default databases (pathway results are provided in Additional file 6: Table S46). Despite using additional taxonomic information, HUMAnN2 was not able to detect any significant shifts in pathways related to tryptophan metabolism (a clinically established hall-mark of PD). The differentially abundant pathways

identified by HUMAnN2 were largely related to the broad category of purine and pyrimidine metabolism which is non-specific to Parkinsonism. It detected a downward shift in some vitamin B and phospholipid metabolism pathways which might be associated with Parkinson's disease.

## Discussion
While the rapid advancement in sequencing technologies has helped researchers resolve the taxonomic diversity of microbial "dark matter" to a great extent, much of its functional diversity remains uncharacterized [18, 38, 58]. Even for the minimal bacterial genome designed by Hutchison et al. [59], the function of one third of the genes could not be determined. Thus, functional annotation remains a difficult task even for well-studied genomes, and it is unsurprising that the sensitivity of all relevant methods is low across the board. Potential reasons why reads often cannot be mapped to functional labels include unknown functionality, non-metabolic functionality, lack of coverage in reference databases, or a non-prokaryotic origin. It is possible to use a much larger off-the-shelf protein database containing computationally predicted functional labels, but doing so is not always advisable because incorporating such databases can increase the chance of erroneous transfer of spurious annotations [31, 60].

More than simply providing an alternative functional profiling tool, Carnelian is able to capture hidden microbial metabolic functional diversity from whole metagenome sequencing reads through its use of a gapped *k*-mer classifier. Being able to accurately label additional ECs manifests partially as an increase in Carnelian's sensitivity. Additional sensitivity alone is suspect, due to the possibility of spurious labels, but we believe that our stricter criteria for database inclusion, combined with training negative examples to reduce false positives, contributed significantly to Carnelian being able to assign a functional label to unknown proteins while minimizing false positives. This ability makes Carnelian a potential tool for annotating novel microbial proteins that are increasingly becoming available [61]. Also, it might be partially due to this ability that, unlike existing methods, Carnelian is able to create functional profiles that are comparable across populations. In multiple large-scale comparative experiments, Carnelian uncovers shared and novel functional similarities and differences across diverse populations and environmental conditions that would go unseen when using existing tools, which are often implicitly designed around taxonomic profiling.

Carnelian detected a high degree of similarity in core metabolic pathways between healthy guts in industrialized and non-industrialized communities, despite significant taxonomic differences [38, 54, 55]. This result is notable given the differences in external pressures (e.g., diet, lifestyle, exposure to toxins) and may indicate the adaptive nature of the gut microbiome. Indeed, many of the enzyme-level variations we found did suggest an adaptive response to industrialized vs non-industrialized dietary differences in carbohydrates (simple sugars vs complex monosaccharides) and proteins (protein-rich vs protein-deficient); this finding agrees with earlier studies [38, 55]. By using different enzymes involved in core metabolic pathways, the healthy guts in these communities can better maintain the overall balance in core metabolic functionality.

We did observe differential read abundance in several xenobiotics metabolism pathways between industrialized and non-industrialized microbiomes (Additional file 5: Table S47). For example, non-industrialized microbiomes showed enrichment of reads in antibiotic resistance ECs and pathways (e.g., beta lactamase, drug metabolism by cytochrome P450). On the other hand, we observed higher read abundance in lipoic acid metabolism, xenobiotics metabolism by cytochrome P450, and phenylpropanoid biosynthesis pathways in the industrialized gut. These findings agree with earlier studies [38, 54, 55]. A potential line of future inquiry would be to investigate these similarities and differences with much larger sample sizes but such is beyond the scope of this study.

Our results with Carnelian indicate concordant dysbiosis in several microbial carbohydrate metabolism pathways in both Chinese and European cohorts for type 2 diabetes. Though existing methods identified variable read abundances in several carbohydrate metabolism pathways, they did not find any common pathways which were statistically significant in both the cohorts. T2D patient guts were found to have higher read abundance in the oxidative phosphorylation pathway, suggesting a higher degree of bacterial defense against oxidative stress and a greater energy imbalance in the patient gut [8, 9]. While the shared dysbiosis in vitamin B metabolism pathways might not be directly related to the disease process, it could be a side-effect of prolonged metformin use by T2D patients in both cohorts [8, 62].

In Crohn's disease case-control cohorts from the USA and Sweden, Carnelian uncovered reduced functional potential of several specific carbohydrate metabolism and amino acid biosynthesis pathways; other tools did not find any concordant dysbiosis. Our results make sense given that microbial carbohydrate metabolism, amino acid synthesis, and selenocompound metabolism pathways were already known to be associated with Crohn's disease [63, 64]. Valine, leucine, and isoleucine have anti-inflammatory roles and are required for intestinal growth and maintenance of mucosal integrity and barrier function;

dietary amino acids have been found to be beneficial for inflammatory bowel disease (IBD) animal models [65]. Additionally, dysbiosis in the microbial biosynthesis of N-glycan can affect the intestinal health of CD patients [66].

For Parkinson's disease (PD), Carnelian's results indicate a downward shift in the gut microbial capacity to synthesize tryptophan, which was not found by mifaser or HUMAnN2 (both the translated search and the full out-of-the-box pipeline). Microbial tryptophan metabolism has been associated with a number of diseases [67], and in particular for Parkinson's, this might affect serotonin production in the host as tryptophan is a known precursor of serotonin. We also found microbial carbohydrate metabolism to be altered in Parkinson's disease which might be a contributor to the insulin impairment observed commonly in Parkinson's patients [68]; glucagon-like peptide-1 receptor agonists, which act in the gut-brain axis pathway and regulate blood glucose, have shown therapeutic potential in clinical studies of PD [69].

Of course, though we find significant alteration in functional capacity of these microbial metabolic pathways, these diseases cannot be characterized by these shifts alone. Integrative approaches involving metabolomics, metagenomics, and metatranscriptomics will likely be required to establish causal relationships between microbial pathways and disease processes in the host. Since disease-associated shifts can often be confounded by antibiotics and other drug usage by participants in a case-control study, the results must be interpreted carefully. Despite these challenges, we were able to show that it is possible to find concordant functional trends across geographically separated case-control cohorts. Our study opens the door to a future where bioprospecting efforts using natural microbes, genetically engineered bacteria, or microbial products targeting specific metabolic pathways in a broad therapeutic context may become possible.

## Conclusion

We have presented here a full pipeline for whole metagenome comparative studies. By integrating together more tailored database curation, probabilistic gene finding, alignment-free functional metagenomic binning, abundance estimation, and the appropriate statistical tools, we show that on a variety of datasets, our tool provides a more comprehensive picture of the functional relatedness of healthy and disease microbiomes than cannot be achieved using existing tools, which implicitly rely on taxonomic binning. We note that while there is an important role for taxonomic binning—indeed, the authors have also developed software for that problem [42]—we believe it essential to be able to perform

comparative studies focused primarily on function, as Carnelian does. Carnelian's modular design enables flexibly running each step of the pipeline independently—for instance, it can be run on either raw sequencing reads (default) or transcriptomic sequences (by bypassing the ORF detection phase). Alternately, should a user prefer to employ other functional profiling tools instead of Carnelian, other components of our pipeline, such as the database curation and statistical tests, may still be of use.

To demonstrate the usefulness of our pipeline, we also analyze a variety of datasets, some publicly available and some newly collected. For type 2 diabetes and Crohn's disease, earlier studies showed only a moderate degree of taxonomic dysbiosis, which did not generalize across different geographic cohorts. With Carnelian, we newly identify concordant changes in the functional capacity of 13 metabolic pathways in European and Chinese type 2 diabetes cohorts and 8 metabolic pathways in US and Swedish Crohn's disease cohorts. Moreover, Carnelian was able to identify several clinically established hallmarks of Parkinson's disease that were not found by other state-of-the-art functional annotation tools. Carnelian-identified EC terms can be used to classify patients and controls with high accuracy. In healthy microbiomes from industrialized and non-industrialized communities, Carnelian identified more functional diversity at both the EC and pathway levels compared to other methods and revealed a high degree of pathway-level similarity in core metabolic functionality.

Carnelian's unique ability to find functional relatedness in diverse metagenomic datasets at the scale of hundreds of samples opens the door to more comprehensive comparative functional metagenomic studies across different geographies, environmental conditions, and time points. We expect Carnelian to be an essential component of the metagenomic analysis toolkit, especially when cross-population comparisons are performed.

## Methods
### Overview of the pipeline

We present here a full pipeline for whole metagenome comparative studies. Our pipeline combines more tailored database curation, probabilistic gene finding, alignment-free functional metagenomic binning, abundance estimation, and appropriate statistical tools for performing comparative functional metagenomics. Figure 1 depicts the main components of our pipeline. The heart of our pipeline is a new compositional tool for functional metagenomic binning called Carnelian. It incorporates probabilistic ORF finding with a compositional gapped classifier ensemble to bin reads into different Enzyme Commission (EC) groups according to their gene content (if any).

Carnelian represents gold standard proteins with complete EC labels in a low-dimensional compact feature space by leveraging Opal-Gallager hashes [42, 43]. These features are then used to train an ensemble of one-against-all classifiers (support vector machines). We implemented the classifier ensemble using the Vowpal-Wabbit (v8.1.1) framework [70, 71]. Negative examples were generated using the "shuffle" program from the HMMER package [47]. The classifiers are trained in an online fashion (one example in memory at a time) using stochastic gradient descent (SGD). The online training capability makes incremental training of Carnelian easy as new verified EC annotations for proteins become available. For more details of the parameters of the classifier ensemble, see Additional file 1: Note S7.

To functionally profile WMS reads, Carnelian first uses FragGeneScan [46] to detect the best possible ORFs from them. FragGeneScan is a unified hidden Markov model framework that incorporates codon usage bias and sequencing error models to probabilistically detect the coding part(s) of the reads. As part of our pipeline, FragGeneScan is run with "short reads" option, because our input is short WMS reads. Since the average substitution error rate for Illumina sequencing is $\sim 0.1\%$, we used the "complete" option with FragGeneScan which assumes 0% error rate. The ORFs predicted by FragGeneScan are encoded into the same compact feature space as in training using Opal-Gallager hashing. Carnelian employs the trained classifier ensemble to bin the feature vectors of the ORFs by EC labels.

All else being equal, the more abundant proteins from an EC label in the microbial sample is, the more reads from them are likely to be sequenced. Therefore, read counts can be used as a proxy for EC abundance in the sample. However, in practice "all else" are never equal. Hence, we borrow intuition from transcriptomics and have Carnelian construct a functional vector by normalizing the read counts as follows:

$$\text{Effective protein length in EC bin } b, \; e_b = p_b - \frac{rl}{3} + 1$$

$$\text{Abundance of EC bin } b, \; \rho_b = \frac{\frac{r_b}{e_b} \times 10^6}{\sum_b \frac{r_b}{e_b}}$$

Here, $p_b$ is the effective protein length (in amino acids) of EC bin $b$ and $rl$ is the average read length (in base pairs). This takes into account the effect of effective protein length of an EC bin as well as the lengths of the proteins in other EC bins while calculating the relative abundance of an EC label in a sample. This normalization further ensures that the relative abundances of the ECs sum up to the same amount in every microbial sample putting the abundances on the same scale. This makes the proportions directly comparable across samples. See Additional file 1: Note S2 for a demonstration of the lack of comparability when no normalization or only RPK normalization is performed; of course, Carnelian also makes the raw counts available should the user desire to experiment with other normalization techniques, but we use this particular normalization for all cross-population comparisons in this study.

## Database curation

We built our gold standard reference dataset by first collecting reviewed prokaryotic proteins from UniProtKB/Swiss-Prot (Feb. 2018) [72, 73] that had both experimental evidence of existence at either the protein or the transcriptomic level and complete EC Numbers associated—EC numbers act as the primary identifiers for metabolic pathway members. We excluded any protein that had computationally inferred functional labels (e.g., by homology), an incomplete EC label, or multiple EC annotations. Indeed, some proteins can have multiple functions. However, these proteins primarily act as enzymes and the secondary functions are mainly non-enzymatic. Therefore, we can safely assume that a protein will have a unique EC label in the reference database. We also collected prokaryotic catalytic residues with complete EC numbers for which a literature reference existed from the Catalytic Site Atlas. We combined these two sets and removed any redundant sequences, which gave us a reference dataset, *EC-2010-DB*, consisting of 7884 proteins with 2010 unique EC numbers (both the dataset and a pre-trained model to bin reads into EC labels are available on the Carnelian's website [74]). Amino acid sequences for these proteins were downloaded from UniProt [72]. This database is designed for profiling the metabolic functional capacity of the microbiome and more suited for cross-comparing healthy and disease microbiomes. Additionally, we provide a database of 1,785,722 proteins from 3285 COG categories and a pre-trained model to classify reads into COG categories on our website, which can be used for microbial functional profiling beyond metabolism.

## Constructing compact feature vectors using Opal-Gallager hashes

Let us consider a sequence fragment of $l$ amino acids, $s \in \Sigma^l$, where $\Sigma = $ standard amino acid alphabet ($|\Sigma| = 20$). A $k$-mer, with $k < l$, is a short word of $k$ contiguous amino acids. Similar to the bag-of-words representation of a document, we define a $k$-mer profile of a sequence $s$ as a vector $f_k(s) \in \mathbb{R}^{20^k}$. We index each $k$-mer with an integer $i$, where $0 \leq i \leq 20^k$ which can be represented by a binary string of length $5k$. Each entry $f_k(s, i) \in f_k(s)$ stores the frequency of the $i$th $k$-mer. Thus, an amino acid fragment of length $l$ can be represented using $k$-mers in $O(20^k)$ space instead of a vector of $O(20^l)$. Using random locality sensitive hash (LSH) functions, we can create

$k$-mer profiles that specify spaced subsequences, rather than contiguous subsequences of fragment $s$. More specifically, we define a random hash function, $h : \Sigma^k \rightarrow \Sigma^r$ to generate a spaced $(k, r)$-mer such that a hashed $k$-mer can be represented by a binary vector of $O(20^r)$ dimensions with corresponding positions set to 1. Here, $r$ denotes the number of positions selected within a $k$-mer window. With this family of LSH functions, we can randomly sample a set of $m$ LSH functions and concatenate them together to represent a $k$-mer profile of a sequence by only $O(m20^r) \ll O(20^k)$ space. However, $k$-mer profiles built with uniformly random LSH functions often have uneven coverage of positions in a sequence unless a large number of such functions are used. To evenly cover positions using a small number $(m)$ of LSH functions, we build upon Opal's modified Gallager design algorithm [42]. Figure S7 in the Additional file 4 depicts an example of how even coverage LSH functions are generated for an amino acid $k$-mer. We used a $k = 8$ and $r = 4$ for the purpose of this paper. More details on the choice of $k$-mer length can be found in Additional file 1: Note S8.

### Benchmarking experiments

We benchmarked our compositional functional profiler, Carnelian against state-of-the-art alignment-based tools, mi-faser and HUMAnN2, and a fast alignment-free tool, Kraken2, using our gold standard database, EC-2010-DB on a number of synthetic metagenomes. Off-the-shelf HUMAnN2 and Kraken2 use taxonomic information in addition to translated searches; to ensure fair comparison, we used only their "translated-search" or "protein-search" mode. All comparisons were based on the EC terms identified by each method using the same gold standard reference database. That is to say, the reference databases we used for the mi-faser and HUMAnN2 and the Kraken2 reference indexes were created with Carnelian's gold standard reference database for unbiased comparison. Detailed performance benchmarks for Carnelian against mi-faser, HUMAnN2, and Kraken2 are available in Additional file 1: Note S5. The exact commands used for running mi-faser, HUMAnN2, and Kraken2 are given in Additional file 1: Note S9 and scripts are available on the Carnelian's website [74].

### Functional profiling of real datasets

We explored two large-scale type 2 diabetes (T2D) studies, two Crohn's disease (CD) studies, and a Parkinson's disease (PD) study for investigating functional dysbiosis in disease vs healthy microbiomes. We analyzed whole metagenome sequencing data from fecal samples of 347 individuals from a Chinese T2D study cohort [8]. Raw paired-end Illumina reads were downloaded from the NCBI short read archive (SRA) (Study accession: SRP008047). We labeled this dataset T2D-Qin.

Additionally, we analyzed fecal metagenome sequencing data from a T2D study performed on a European cohort of 145 women with either T2D or impaired glucose tolerance (IGT) or normal glucose tolerance (NGT) [9]. Since we aimed at finding the differences in microbial metabolic function between T2D patients and healthy individuals, we did not include the IGT individuals in our analysis. We downloaded publicly available raw Illumina HiSeq 2000 paired-end reads from NCBI SRA (Study accession: ERP002469); each individual metagenome contained $\sim$ 3 Gb on average. We labeled this dataset T2D-Karlsson. We further analyzed two Crohn's disease case-control datasets: 53 US individuals from HMP pilot phase and 62 Swedish individuals from a Swedish cohort [1]. We downloaded publicly available raw Illumina HiSeq 2000 paired-end reads for the US cohort (CD-HMP dataset) from the IBDMDB website [52]. Raw reads for the Swedish cohort (CD-Swedish) were downloaded from NCBI SRA (Study accession: SRP002423). We also analyzed whole metagenome sequencing reads from the fecal samples of 20 patients and 21 healthy individuals in an early stage L-DOPA naïve PD case-control study [57]. All the participants in the study were male and age-matched. We downloaded publicly available raw Illumina HiSeq 2500 paired-end reads from NCBI SRA (Study accession: ERP019674). We labeled this dataset PD-Bedarf. Metadata of the samples from each study are provided in Additional file 7: Tables S48-S52.

For investigating the functional relatedness of the healthy microbiomes in industrialized and non-industrialized communities, we analyzed gut microbiomes of four cohorts (84 individuals from Boston, 35 Baka individuals from Cameroon, 50 individuals from the Gimbichu region in Ethiopia, and 112 individuals from Madagascar of Betsimisaraka and Tsimihety ethnicity). The Baka dataset is unpublished data on a sensitive indigenous population from the Eric Alm lab—see the Carnelian website for the accession numbers once they are available. The datasets from Boston, Ethiopia, and Madagascar were contributed by two recent studies [38, 44] and are publicly available at NCBI's SRA with study accessions: SRP200548, SRP168387, and SRP156699. Metadata of the samples from each study are given in Additional file 7: Tables S53–S56.

### Preprocessing steps for raw reads

We used Trimmomatic v0.36 [75] for adapter trimming and quality filtering with a quality threshold of 30 and a minimum length of 60 bp (paired-end mode for Illumina reads and single-end mode for Roche 454 reads). DeconSeq v0.4.3 [76] was used to remove contaminating human sequences with the human reference genome GRCh38 as the database. For paired-end reads, we kept only the read-pairs for which both sequences survived quality control.

These steps were applied to all the datasets. In the T2D-Qin dataset, 241 of the samples survived the preprocessing step and were used for subsequent analyses.

### Quantifying microbial functional variation in real datasets

Carnelian outputs the effective read counts per EC label (i.e., normalized read counts against effective protein length per EC bin and a per million scaling factor) as abundance estimates. For the other three methods, we applied the same normalization on the raw read counts produced by them to ensure an unbiased comparison. Pathway abundances were calculated by grouping the ECs into KEGG metabolic pathways and summing the effective read counts. Pathway coverage was calculated as the ratio of the number of mapped ECs identified by a method to the total number of reference ECs present in the pathway.

For the studies with two groups of microbiomes (case vs control, industrialized vs non-industrialized), we created an effective counts matrix using Carnelian generated functional profiles and performed pairwise Wilcoxon rank-sum test (Mann-Whitney $U$ test). A Benjamini-Hochberg (BH) false discovery rate (FDR)-corrected $p$ value threshold of 0.05 was used as a test of significance. Additional log-fold-change thresholds have been selected for each dataset (mentioned in the main text).

To determine the significance of the common pathways between geographically separated disease cohorts, we combined the individual $p$ values per pathway from different studies of the same disease using Fisher's combined probability test (Fig. 1: green). To investigate the co-abundance of microbial metabolic pathways between healthy microbiomes of industrialized and non-industrialized communities, we computed Kendall's rank correlation of the pathway abundance profiles of the two groups. Next, we performed Ward-linkage hierarchical clustering using Euclidean distance on the pathway co-abundance matrix (correlation matrix). To determine whether the centroids and dispersion of the pathway clusters are significantly different between the non-industrialized and industrialized microbiomes, permutational multivariate analysis of variance (PERMANOVA) test was performed using "adonis" function available through the "vegan" package in R (Fig. 1: blue). For measuring functional diversity in a sample, we calculated the Shannon-Wiener diversity index of the EC and pathway profiles of the samples using the "vegan" package available in R.

### Availability of Carnelian

Carnelian is open source and freely licensed (MIT License). Source code of Carnelian is available at our website [74], Github [77], and Zenodo [78].

## Supplementary information

---

**Additional file 1:** Supplementary Notes. Contains Supplementary Notes S1–S9.

**Additional file 2:** Results from Type-2 Diabetes Cohorts. Contains Supplementary Tables S1–S16.

**Additional file 3:** Results from Crohn's Disease Cohorts. Contains Supplementary Tables S17–S32.

**Additional file 4:** Supplementary Figures. Contains Supplementary Figures S1–S7.

**Additional file 5:** Results from Case-Control Classification and Functional Relatedness of Healthy Microbiomes. Contains Supplementary Tables S33–S37 and S47.

**Additional file 6:** Results from Parkinson's Disease Cohort. Contains Supplementary Tables S38–S46.

**Additional file 7:** Metadata of Study Cohorts. Contains Supplementary Tables S48–S56.

**Additional file 8:** Review history.

---

#### Peer review information

#### Acknowledgements
We are incredibly grateful to Eric Alm, Mathieu Groussin, and Mathilde Poyet for sharing the gut microbiome data from Bostonian and Baka individuals from the Global Microbiome Conservancy project (http://microbiomeconservancy.org/) with us. We also greatly appreciate the useful discussions we had with them on the functional relatedness of industrialized and non-industrialized individuals. We further thank Vicki Mountain and Hera Vlamakis for pointing us to the IBDMDB dataset for Crohn's disease. Finally, we thank Hyunghoon Cho, Craig Mak, and Ariya Shajii for useful discussions and comments.

#### Review history
The review history is available as Additional file 8.

#### Authors' contributions
SN and BB conceived the idea and designed the experiments. SN implemented the Carnelian pipeline and performed the experiments. SN, YWY, and BB analyzed the results and wrote the manuscript. BB guided the research. All authors read and approved the final manuscript.

#### Funding
S.N. gratefully acknowledges support from the International Fulbright Science and Technology Fellowship and the Ludwig Center for Molecular Oncology Graduate Fellowship. S.N. and B.B. are partially supported by the National Institutes of Health (NIH) R01GM01108348 and Center for Microbiome Informatics and Therapeutics at the Broad Institute. Y.W.Y. is supported by the NIH NLM training grant T15LM007092. This content is solely the responsibility of the authors and does not reflect the official views of the funding authorities.

#### Availability of data and materials
All the datasets analyzed in this study are listed below along with the information on their availability. Additionally, Carnelian's source code is freely available from Github [77] and Zenodo [78] under the MIT license.

- EC-2010-DB: Carnelian's gold standard dataset of prokaryotic proteins annotated with verified Enzyme Commission (EC) numbers. Available on the Carnelian's website [74].
- T2D-Qin: Metagenomic data from the Chinese type-2 diabetes case-control study. Available from the NCBI's SRA with the study accession number SRP008047 [79]. Metadata of the dataset is provided in Additional file 7: Table S48.
- T2D-Karlsson: Metagenomic data from the European type 2 diabetes study. Available from the EMBL-EBI's ENA with the study accession number ERP002469 [80]. Metadata of the dataset is provided in Additional file 7: Table S49.

- CD-HMP: Metagenomic data from Crohn's disease case-control cohort consisting of US individuals. Available from the IBDMDB website [52]. Metadata of the dataset is provided in Additional file 7: Table S50.
- CD-Swedish: Metagenomic data from the Swedish Crohn's disease case-control cohort. Available from the NCBI's SRA with the study accession number SRP002423 [81]. Metadata of the dataset is provided in Additional file 7: Table S51.
- PD-Bedarf: Metagenomic data from the Parkinson's disease case-control study. Available from the EMBL-EBI's ENA with the study accession number ERP019674 [82]. Metadata of the dataset is provided in Additional file 7: Table S52.
- Boston-Industrialized: Metagenomic data from healthy industrialized individuals from Boston. Available from the NCBI's SRA with the study accession number SRP200548 [83]. Metadata of the dataset is provided in Additional file 7: Table S53.
- Cameroon-Non_Industrialized: Unpublished metagenomic data from healthy individuals of Baka Ethnicity from Cameroon. Currently available from the Alm lab (https://almlab.mit.edu). Metadata of the dataset is provided in Additional file 7: Table S54.
- Ethiopia-Non_Industrialized: Metagenomic data from healthy Ethiopian individuals from the Gimbichu region. Available from the NCBI's SRA with the study accession number SRP168387 [84]. Metadata of the dataset is provided in Additional file 7: Table S55.
- Madagascar-Non_Industrialized: Metagenomic data from healthy individuals of Betsimisaraka and Tsimihety ethnicity from Madagascar. Available from the NCBI's SRA with the study accession number SRP156699 [85]. Metadata of the dataset is provided in Additional file 7: Table S56.

### Ethics approval and consent to participate
Most of the results in this paper are based on publicly available whole metagenome sequencing read data from NCBI's SRA, EMBL-EBI's ENA, and JGI. Use of these data does not require ethics approval. The unpublished data from the Eric Alm lab (MIT) was collected with proper ethics approval and consents from participants and is available upon request.

### Competing interests
The authors declare that they have no competing interests.

### Author details
[1]Computer Science and Artificial Intelligence Laboratory, MIT, 77 Massachusetts Ave, MA 02139, Cambridge, USA. [2]Department of Mathematics, University of Toronto, ON, M5S 2E4 Toronto, Canada. [3]Department of Computer and Mathematical Sciences, University of Toronto at Scarborough, ON, M1C 1A4 Toronto, Canada. [4]Department of Mathematics, MIT, 77 Massachusetts Ave, MA 02139 Cambridge, USA.

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

## 
