## [**Additional file 8** Review history. · Genome Biology]

Review History

First round of review

Reviewer 1

Are you able to assess all statistics in the manuscript, including the appropriateness of statistical tests used? Yes, and I have assessed the statistics in my report.

Comments to author:

Summary

In this manuscript the authors described a novel computational pipeline, Carnelian, for metabolic functional profiling of whole metagenome sequencing (WMS) data. This pipeline is composed of multiple elements: an algorithm for rapid protein sequence matching based on k-mer profiles, a reference database with EC terms as basic functional units, and protocols for predicting ORFs, normalizing profiles to cross-sample comparison, etc. The authors benchmarked Carnelian against functional profilers like HUMAnN2, Kraken2 and mi-faser and observed higher sensitivity in most cases, along with acceptable precision and computational expense. They tested Carnelian on real metagenome datasets from three cohorts: Type 2 diabetes (T2D), Crohn's disease (CD) and Parkinson's disease (PD), and showed that certain functional features of potential importance can only be detected by Carnelian but not by existing tools. They made Carnelian and its source code publicly available.

The study is impressively comprehensive, with very dense materials, addressing multiple aspects that bioinformaticians and end users would be interested in. The author made great efforts in attempt of providing an all-in-one solution to a complex, multi-step bioinformatics question, which I highly appreciate.

However, I have multiple major concerns over the theory, design, results and interpretations of Carnelian, making me overall highly skeptical of the claimed value of this work. My critiques are detailed below.

Design

Sensitivity

Throughout the manuscript, the metric "sensitivity" has been repeatedly mentioned as an advantage of Carnelian over existing tools. Examples: "...lack of sensitivity of existing functional annotation tools..." (page 2 line 13), "HUMAnN2 lacks sensitivity..." (page 2 line 32), "compositional (k-mer based) taxonomic binning methods have been shown to achieve higher sensitivity compared to read-alignment approaches..." (page 2 line 34). The ability of Carnelian in detecting things "from non-annotated species" was also repeatedly mentioned. The text overall delivers an impression that every gene should have a function, and the function should be identifiable. In another word, "the more the better". This is dangerous.

A large proportion of proteins are of unknown functions. This is not because existing protein function predicting tools are not sensitive enough (as the authors implicated), but because people just haven't been able to discover their functional roles. For an extreme example, the minimal bacterial genome generated by Hutchison et al., Science, 2016, has only 473 genes, in which 149 are of unknown function. That is, even for the (probably) most basic life form people can ever reach, people don't know the function of one third of its genes. Therefore, one shouldn't be surprised that in most publicly available microbial genomes there are large proportions of "hypothetical proteins", and probably shouldn't try to create labels for them

without using a pipettor.

Meanwhile, assigning reads to a category without knowing whether it is correct is meaningless. If one wants, one can specify an e-value cutoff = 100 and get very high sensitivity with an old-school BLAST search, but this shouldn't be considered as a major breakthrough (also see my comments on precision below).

Page 8 line 29: "Of course, some reads could not be mapped by any method, including ours." Of course a large proportion of reads cannot be mapped. I will be very skeptical if most reads are actually mapped. What are the proportions of reads mapped in the three real datasets?

Taxonomic assignment

As the authors admitted (page 11 line 60), HUMAnN2 and Kraken2 use taxonomic information, whereas Carnelian cannot. This is a very important feature for modern functional profilers (but see comment on Kraken below). The authors disabled them when doing benchmarking, which is unfair.

All the analysis on pathway coverage in Carnelian is based on the assumption that all observed genes function together--but in reality, genes are separated in discrete chambers--microbial cells, and only genes within the same cell constitute pathways, except for a few highly mutualistic, obligatory situations. That's why it is of importance to assess this information when performing functional profiling. While precisely assigning functional genes to host genomes requires assembly and binning, modern read-based profilers resort to performing taxonomic and functional profiling together. For example, HUMAnN2 can report "stratified" and non-stratified profiles. It seems that Carnelian lacks this important design, which renders its outcome less meaningful.

Cross-sample comparability

The authors claimed that one major advantage of Carnelian is that results can be compared cross samples. They detailed their design in page 3 lines 49-60. The main change over existing tools is that they normalized hit numbers by "effective protein length", the equation of which is provided in page 7 line 11. However, I do not see the justification why this metric is superior to no normalization. I can imagine that this equation is to provide correction given the short nature of Illumina sequencing reads versus the long nature of whole proteins in the reference database. But the authors need to provide explanation, as well as evidence in both theory and benchmark tests to demonstrate why it is better and how it represents a significant innovation.

Actually, when I first see the claim of cross-sample comparability in the abstract, I was expecting to see things like ranking, log-ratio transform, etc, which have been widely experimented by recent researchers. The current treatment in Carnelian is quite simple and I am not convinced that it represents a major innovation.

Multiple assignment

Page 11 line 23: "We excluded any protein that had computationally inferred functional labels (e.g. by homology), an incomplete EC label, or multiple EC annotations".

It is very common for a protein to possess multiple functions--how does Carnelian deal with this situation? If each protein is assigned to only one EC, I would be very unconvinced by any claims of the goodness of results.

ORF finding

The feature "probabilistic ORF finding" was emphasized multiple times including the abstract. But wait, is it just FragGeneScan, a simple tool published 10 years ago and already widely-used? I am not saying that the authors cannot describe FragGeneScan as "probabilistic ORF finding". I just think it is not worth stating it in such way that it sounds like an innovation made by the authors.

Reliance on EC terms

The basic units of the Carnelian reference database are enzyme commission (EC) terms. EC terms are commonly used and are a nice feature in many functional annotation tools.

However, not all proteins are enzymes. Reporting EC terms only is a limitation in many use cases. Other protein function catalogs are usually more comprehensive. For example, gene ontology (GO) terms have three top-level categories: "cellular component", "molecular function", and "biological process".

The authors noticed this and described Carnelian as "focused on metabolic functionality", "as opposed to using typical protein database..." (page 2 line 47). This is accurate, however I do not see why this is an advantage instead of a limitation ("(Carnelian) is better suited for..."page 2 line 43). This needs to be justified.

Benchmark

Precision

The authors did present precisions, F1 scores (Note X3) and AUCs (Fig. 2). Why didn't the authors also emphasize on them in the main text? One possible reason I can imagine is that Carnelian's precision is not as high as other tools (especially HUMAnN2, e.g., Fig. FX4, Table TX8-9). Of course, the difference isn't dramatic, but meanwhile the increase in sensitivity isn't dramatic from those tables & figures either.

This result is (partially) in my expectation, since HUMAnN2 is based on amino acid sequence alignment (and a database of marker genes instead of whole genomes, but here the authors used a uniform database), being accurate at the cost of lower sensitivity, whereas Kraken is known suffering higher false positive assignment due to the nature of k-mer matching. But it is bit surprising that Kraken2 performed worse in both precision and recall according to Fig. FX4.

Database

For a fair comparison, the authors used the same reference database to benchmark all four tools. This practice is correct and should be encouraged. However, therefore the analysis shown here was actually benchmarking sequence aligners, not functional profilers.

Some dedicated functional profilers, such as HUMAnN2, has its official database or protocol for making database (ChocoPhlAn + UniRef + MetaCyc). Hooking HUMAnN2 to the Carnelian database and showing it is less sensitive than Carnelian seems to be unfair.

And why don't the authors directly benchmark Carnelian against protein sequence aligners, such as BLASTP, DIAMOND, HMMER, etc.?

Kraken

I do not know Kraken1/2 was designed and promoted as a functional profiler. It is a taxonomic profiler. The authors need to explain why they included Kraken in the benchmarks.

And, Kraken features a heuristic based on a taxonomic hierarchy. Does the Kraken database built by the authors also have any sort of hierarchy? If not this is another source of unfairness.

Precise mode

I noticed that Carnelian has a "precise" mode (Note X4). I wonder how precision / sensitivity would be in this mode?

Test dataset

The authors created a synthetic community for benchmarking, as shown in Table TX5. However, a microbiologist will quickly realize that it does not look like a realistic human gut microbial community. To justify this design, the authors stated that they are the 20 most abundant bacterial species in HMP stool samples. I am not clear how this information was drawn. But in Fig. 3a of Huttenhower et al. Nature, 2012 (which the authors cited) indicated otherwise. I suggest that the Carnelian team have a microbiologist reviewing their experiments, if not already.

Statistical test

To test the differential clustering of metagenomic functional profiles by certain metadata categories (e.g., industrialized and non-industrialized), the authors relied on the PERMANOVA test, which is a good choice for this question. However, the interpretation of PERMANOVA result is problematic. The authors report p-values alone. But what is more important for PERMANOVA is the pseudo-F statistic, which indicates the ratio of inter-category variance over intra-category variance; and the R-squared value, which indicates the proportion of variance explained by this metadata category. Both are available from the "adonis" function which the authors used, and should be reported. In contrast, the p-value of a PERMANOVA test is not a good indicator of group separation significance, and is almost always small given the sample size is large. (Strangely, the p-values shown in Fig. 3b seem to be very large.)

Introduction

The authors reviewed the previous and current trends on the topic of shotgun metagenomic data analysis and functional profiling. I noted that many representative studies and tools have been mentioned, which I appreciate. However, I found the overall logic misleading.

Abstract and page 2 line 9: "these attempts have generally been based on targeted 16S rRNA sequencing data..." Come on. This is 2019. A large number of studies already adopted shotgun data, and even multi-omics data.

Another reason suggested by the authors is that existing tools "are limited in their ability to directly compare microbial metabolic function across samples and studies." I have detailed my critique above.

Page 1 line 59: "though quite related to functional profiling..." Bit distracting. What is the main point?

Page 2 line 13: "...lack of sensitivity of existing functional annotation tools, inconsistencies in annotations... across... databases, and lack of comparability of existing... statistics". These critiques are strong, and need to be backed by evidence. Despite the authors showed that in very specific experiments Carnelian may have higher sensitivity, but more are expected in order to make these claims.

Biology

Page 8 line 60: "nonindustrialized microbiomes showed enrichment of reads in antibiotic resistance ECs and pathways (e.g. beta-lactamase...": Why would less industrialized communities have higher abundance of antibiotic resistance genes? This is not intuitive.

Terminology

The authors described Carnelian as a "compositional" tool, and "compositional" means "k-mer based" (page 2 line 39). I find this term ambiguous. First, in traditional genomics, the term "compositional" also describes GC-content, tetramer frequencies, codon usage signature, etc. Second, "compositional" can be easily confused with the statistical term "compositional" (meaning adds up to one) which has been frequently used to characterize microbiome data. Therefore, I would suggest the authors be more specific. Why not just using "k-mer based"?

Formatting

The main text PDF is gigantic. The figures seem to be formatted in an unfavorable way, which slowed down my computer when I was trying to view them. Please fix.

Line numbers are not incremental across pages and are not aligned to each line. Hard to track. Please reformat.

There are two Table TX6's. Indexing problem?

Final Words

Based on the reasons detailed above, I do not agree that "(Carnelian) newly enables using whole metagenome sequencing for meaningful large-scale comparative functional metagenomic studies across diverse populations.". I think this work needs to be substantially reworked before it can be considered for publication.

Reviewer 2

Are you able to assess all statistics in the manuscript, including the appropriateness of statistical tests used? Yes, and I have assessed the statistics in my report.

Comments to author:

In this paper, the author demonstrated an effective method for metabolic functional profiling of metagenomic studies. The Carnelian is a useful tool for meaningful large-scale comparative functional metagenomic studies across diverse populations. Some specific comments and suggestions are as follows:

1. Section "results", Second paragraph, "Thus, we constructed our gold standard reference database with curated prokaryotic proteins that have verified unique and complete EC labels which provide a direct mapping to KEGG metabolic pathways for our later analyses.". How big is your database and how many sequences it contains? What is the difference between this database and the known database, and how many proteins are newly added? Please be more precise in the results.

2. The ORF detection must depend on the parameters of the program, what were the parameters used by the authors, so how robust is this conclusion?

3. Section "Carnelian reveals novel and shared functional dysbiosis across disease studies", paragraph three, "At the pathway level, we found the PD gut to have lower read abundance in several carbohydrate metabolism pathways (BH-corrected Wilcoxon rank sum test p-value < 0.05 and absolute log fold change > 0.11)", what are the specific carbohydrate metabolism pathways? A supplementary table with those pathways may be interesting for the reader.

4. Section "Carnelian reveals novel and shared functional dysbiosis across disease studies", paragraph six, "Carnelian also finds lower abundance of several EC terms including 1.2.1.19, 3.5.1.16, 4.1.1.18, and

5.3.3.10 which play crucial roles in the metabolism of several essential amino acids including arginine, proline, lysine, tyrosine, etc.". the reader may not be expected to know by heart the meaning of all digital terms notations, explanations need to be provided throughout.

5. Is the inconsistency of the metabolic pathways detected in Carnelian, mi-faser, HUMAnN2 and Kraken2 mainly due to the difference in the database on which each software depends, or the difference caused by the algorithm of the software itself?

General Comments to Reviewers:

We have performed the following *additional* experiments:

- To demonstrate Carnelian's enhanced ability to find novel biology in case-control comparisons despite not using any taxonomic information, we ran the full pipeline of state-of-the-art HUMAnN2 with its default databases, including taxonomic information, on the Parkinson's data set. Even after including taxonomic information, HUMAnN2 was unable to find known hallmarks of Parkinson's which Carnelian did.
- We have shown that Carnelian can correctly train functional labels for sequences that are not in the training set. Due to the lack of a gold standard database, we analyzed several known remote homologs using Carnelian and compared our performance against state-of-the-art tools mi-faser, HUMAnN2, and Kraken2.
- To demonstrate Carnelian's superior annotation performance on less-studied populations, we ran full HUMAnN2 (with default databases and taxonomic information) on our novel in-house Baka data set and a subset of our in-house Boston data set. On the Baka data set, HUMAnN2 detects only a few known species per sample, identified fewer enzyme commission (EC) terms, and captured less functional diversity as compared to its results on the industrialized microbiomes of Bostonian individuals, likely because many of the species present have not been characterized yet. Contrastingly, Carnelian detects more EC terms (likely due to its additional labeling of unalignable reads), captures more functional diversity compared to HUMAnN2, and finds the expected functional trends.
- We have explained the intuition behind our proposed normalization through example scenarios and demonstrated through further experiments that our proposed normalization is indeed crucial for comparative profiling.
- As a response to Reviewer 1's comments on databases, we now provide on our website an additional database of 1,785,722 proteins from 3,285 COG categories and a pre-trained model to classify reads into COG categories, which can be used for microbial functional profiling beyond metabolism.

Importantly, we now make clearer in the text that we did indeed benchmark all methods against the same database; we apologize for not making this fact clear to the reviewers in the original submission. Also noteworthy, we did not perform benchmarking separately against protein sequence aligners (e.g. Blastp, PSI-Blast, RapSearch2, Hmmer), because these are included in our comparisons to HUMAnN2 and mi-faser, which already use the state-of-the-art aligner DIAMOND as part of their functional labeling pipeline; DIAMOND is orders of magnitude faster than other protein sequence aligners at comparable precision and sensitivity [Buchfink et al., Nature Methods, 2014].

Towards "additional clarification," we now make sure to place existing experiments in the correct context: for example, the better classification performance using Carnelian ECs implies that the additional labels generated by Carnelian are not spurious, and Carnelian's better

functional concordance across populations for the same disease is evidence for our method's improvements over existing methods in comparative functional profiling.

We especially thank Reviewer 1 for his/her positive comment on the comprehensiveness of our practical pipeline: "The author made great efforts in attempt of providing an all-in-one solution to a complex, multi-step bioinformatics question, which I highly appreciate." Additionally, with respect to the central purpose and metrics for functional profiling, Reviewer 1 *correctly* pointed out that sensitivity is an imperfect proxy for the performance of a functional profiler, writing '*The text overall delivers an impression that every gene should have a function, and the function should be identifiable. In another word, "the more the better". This is dangerous.*' Due to his/her insightful comments about generic sensitivity analysis used in the field, we realised that we needed further proof Carnelian's additional labels are in fact not spurious. We have substantially revised the text to not focus on sensitivity, but instead on our ability to carry out comparative functional profiles *across populations*. We believe this focus better highlights the biological implications of our study.

However, we respectfully disagree that we "probably shouldn't try to create labels for them without using a pipettor" (Reviewer 1). While actual experiments are ideal, they are prohibitively costly, especially for less well-studied populations. Such a restriction would significantly bias our analyses towards the industrialized world, reducing the utility of bioinformatics analysis across populations. Indeed, part of the reason we conferred with the Eric Alm research group on developing Carnelian was to compare the in-house Baka population data set, which our additional experiments demonstrate to not be amenable to analysis through standard tools like HUMAnN2, to an "industrialized" in-house data set from Boston. Notably, his group is incorporating Carnelian into their analysis pipelines, as other methods are insufficient for the large-scale global studies they are conducting.

Please find attached a point-by-point response to each of the reviewer comments (our comments in blue), with more details about additional experiments. Also, we provide a new manuscript with changes highlighted and a DOI link from zenodo for Carnelian's source code, as requested. We would like to express our sincerest appreciation to the editors and referees for their careful reading of our manuscript, which we feel have very much served to improve our presentation and content. We hope the paper is now acceptable for publication in *Genome Biology*.

Point-by-point Responses to Reviewers (in blue):

Reviewer #1: Summary

In this manuscript the authors described a novel computational pipeline, Carnelian, for metabolic functional profiling of whole metagenome sequencing (WMS) data. This pipeline is composed of multiple elements: an algorithm for rapid protein sequence matching based on k-mer profiles, a reference database with EC terms as basic functional units, and protocols for predicting ORFs, normalizing profiles to cross-sample comparison, etc. The authors benchmarked Carnelian against functional profilers like HUMAnN2, Kraken2 and mi-faser and observed higher

sensitivity in most cases, along with acceptable precision and computational expense. They tested Carnelian on real metagenome data sets from three cohorts: Type 2 diabetes (T2D), Crohn's disease (CD) and Parkinson's disease (PD), and showed that certain functional features of potential importance can only be detected by Carnelian but not by existing tools. They made Carnelian and its source code publicly available.

The study is impressively comprehensive, with very dense materials, addressing multiple aspects that bioinformaticians and end users would be interested in. The author made great efforts in attempt of providing an all-in-one solution to a complex, multi-step bioinformatics question, which I highly appreciate.

We really appreciate the positive assessment. In particular, we'd like to emphasize that Carnelian identifies concordant functional changes across geographically separated disease cohorts of T2D and CD which cannot be identified by existing methods nor earlier comparative metagenomics studies. Carnelian also uncovers pathway-level similarities in the core metabolic functional capabilities of the microbiomes of healthy populations with different subsistence strategies--not observed in earlier studies. Furthermore, Carnelian-identified enzyme commission (EC) labels can stratify patients and controls with higher accuracy compared to the ECs identified by other methods.

However, I have multiple major concerns over the theory, design, results and interpretations of Carnelian, making me overall highly skeptical of the claimed value of this work. My critiques are detailed below.

We apologize for not making our intent and contributions clearer and hope we are able to assuage concerns in our responses below and in the revised text. Thank you for your careful review of our presentation and pointing out areas of potential unintended misinterpretation.

Design

Sensitivity

Throughout the manuscript, the metric "sensitivity" has been repeatedly mentioned as an advantage of Carnelian over existing tools. Examples: "...lack of sensitivity of existing functional annotation tools..." (page 2 line 13), "HUMAN2 lacks sensitivity..." (page 2 line 32), "compositional (k-mer based) taxonomic binning methods have been shown to achieve higher sensitivity compared to read-alignment approaches..." (page 2 line 34). The ability of Carnelian in detecting things "from non-annotated species" was also repeatedly mentioned. The text overall delivers an impression that every gene should have a function, and the function should be identifiable. In another word, "the more the better". This is dangerous.

We apologize for having given the incorrect impression that we should be able to identify the function of every gene. Indeed, we specifically train our classifier with an "unknown" label to prevent spuriously forcing assignments of genes to function (see Methods, 2nd paragraph). Due to the reviewer's insightful comments, we realized that we were using "sensitivity" as an

imperfect proxy for Carnelian's strength, which is in classifying function from unknown protein sequences. Instead of using "sensitivity", we now instead give other evidence that Carnelian's additional labels are in fact justified (e.g. the fact that using Carnelian gives better EC terms for classification, or that Carnelian is sometimes able to correctly classify remote homologs that have the same function as a known protein). Though we keep the sensitivity benchmark data, we now add the caveat in the main text that sensitivity is only meaningful when spurious labels are minimized.

A large proportion of proteins are of unknown functions. This is not because existing protein function predicting tools are not sensitive enough (as the authors implicated), but because people just haven't been able to discover their functional roles. For an extreme example, the minimal bacterial genome generated by Hutchison et al., *Science*, 2016, has only 473 genes, in which 149 are of unknown function. That is, even for the (probably) most basic life form people can ever reach, people don't know the function of one third of its genes. Therefore, one shouldn't be surprised that in most publicly available microbial genomes there are large proportions of "hypothetical proteins", and probably shouldn't try to create labels for them without using a pipettor.

While we agree that in an ideal world, label assignments for proteins are done via experimentation---indeed, in our reference database, we specifically curated only the prokaryotic proteins with verified functions---this severely curtails our ability to study novel populations---wet lab experiments characterizing the microbiome of non-industrialized populations are both less common and more expensive, so to properly perform comparative studies using RNA-seq, we need some means of measuring function (even if approximately with some error) that is not as systematically biased against less studied populations. We now discuss this in the fourth paragraph of the revised Background section.

Meanwhile, assigning reads to a category without knowing whether it is correct is meaningless. If one wants, one can specify an e-value cutoff = 100 and get very high sensitivity with an old-school BLAST search, but this shouldn't be considered as a major breakthrough (also see my comments on precision below).

As we discussed above, we agree with the reviewer here, and have corrected our earlier reliance on sensitivity, and use other metrics instead (such as accuracy of classification and finding expected functional shifts based on analysing the biological data).

We should additionally mention though that Carnelian does not arbitrarily assign reads to functional categories and claim to achieve higher sensitivity. Carnelian's classifier ensemble includes randomly shuffled sequences generated via HMMER as negative examples (N.N.N.N label) during training. If a given sequence has an intermediate k-mer based representation that is similar to the learned representation of a functional category in the reference database, only then would Carnelian assign the sequence to this functional category. If a given sequence is not similar to any of the functional categories in the k-mer space, it is labeled N.N.N.N (i.e., no label). Comparing this design with BLAST search with an e-value cutoff = 100 entirely ignores the ability of our classification pipeline to measure its own uncertainty and choose when or when

not to give an EC assignment. We now mention the inclusion of negative examples in the training phase of Carnelian's classifier ensemble in the main text (start of the Results section; 3rd paragraph).

Page 8 line 29: "Of course, some reads could not be mapped by any method, including ours." Of course a large proportion of reads cannot be mapped. I will be very skeptical if most reads are actually mapped. What are the proportions of reads mapped in the three real data sets?

We agree with the reviewer that a large portion of the reads cannot be mapped. In fact, for the real data sets, on average, ~5-15% of reads were mapped to the EC database by all four methods. It was a mistake on our part to write "some" instead of "many," and we are grateful to the reviewer for pointing that out. We have updated the discussion to deliver our intended meaning (the revised first three paragraphs of the Discussion section).

Taxonomic assignment

As the authors admitted (page 11 line 60), HUMAnN2 and Kraken2 use taxonomic information, whereas Carnelian cannot. This is a very important feature for modern functional profilers (but see comment on Kraken below). The authors disabled them when doing benchmarking, which is unfair.

We agree that it is not an apples to apples comparison. The reason for this choice is due to the design goals of Carnelian. As discussed above in our response about label assignments, we are particularly interested in developing tools for populations whose microbiomes are not fully characterized, and whose bacterial strains may not already be in our databases. The taxonomic information we have available is incomplete for those novel data sets, and so in all of our benchmarking, which is generally on data sets where our taxonomic information is quite good, we chose to use only the translated searches of HUMAnN2 and Kraken2 which do not use any taxonomic information.

However, given the reviewer's criticisms, we ran an additional experiment using HUMAnN2 on the Parkinson's Disease data set, where we use its full recommended pipeline, including its own reference database and full taxonomic information. Even with taxonomic information, HUMAnN2 did not perform significantly better in finding Parkinson's specific pathways. Thus, we do not believe that restricting taxonomic information severely impacted our benchmarks. See the fourth subsection in the Results section as well as Supplementary Table T46 in Additional File 6.

All the analysis on pathway coverage in Carnelian is based on the assumption that all observed genes function together--but in reality, genes are separated in discrete chambers--microbial cells, and only genes within the same cell constitute pathways, except for a few highly mutualistic, obligatory situations. That's why it is of importance to assess this information when performing functional profiling. While precisely assigning functional genes to host genomes requires assembly and binning, modern read-based profilers resort to performing taxonomic and functional profiling together. For example, HUMAnN2 can report "stratified" and non-stratified

profiles. It seems that Carnelian lacks this important design, which renders its outcome less meaningful.

We agree with the reviewer that genes act within individual microbial entities and it would definitely be helpful to have pathway profiles stratified by taxonomic units. However, constructing stratified EC- and pathway-profiles from short reads is an extremely challenging problem considering how common horizontal gene transfers are [Smillie et al., Nature, 2011]. The reviewer is right in pointing out that ideally it would require assembling species genomes from the reads and then infer both species identity and gene contents simultaneously. But this is difficult too, as assembly is a slow and lossy process and assemblies often are incomplete and contain errors. So claiming the presence/absence of genes and ECs with such assemblies can be a stretch.

HUMAnN2's treatment of this complex problem is rather simplistic. It looks for marker genes of known species in the short reads and then aligns the reads to identified species genomes using MetaPhlAn2. It performs an additional translated search on the unmapped reads from the previous step using DIAMOND. Current public repositories of microbial genomes are not comprehensive. So, often a large portion of the genes and pathways in HUMAnN2-generated profiles correspond to uncategorized things. A recent large scale study looking at ~9k genomes spanning different body sites, ages, countries, and lifestyles found that more than 77% assembled read clusters did not correspond to any genomes available in public repositories [Pasolli et al., Cell 2019]. The authors focused their discussion on the unstratified functional profiles when comparing different age groups, lifestyles, etc. [Table S5 in Pasolli et al., Cell 2019] even though they used full HUMAnN2 pipeline in their analysis.

Indeed, when analyzing data from less studied populations---so often the case in metagenomic analysis, a large fraction of reads sequenced do not directly correspond to proteins of known species [Lloyd et al., 2016; Pasolli et al., 2019]. As a result, methods that depend on alignment do not perform as well. We observe this problem when studying our novel in-house data set from the non-industrialized Baka population. We ran the full HUMAnN2 pipeline on WMS reads from Baka individuals. Even after using taxonomic information from the entire ChocoPhlAn database, HUMAnN2 could map ~ 10% reads and identified ~ 30 species per sample (third subsection in the updated Results section and Supplementary Note X6 in Additional File 1). It captured less functional diversity in Baka microbiomes as compared to the industrialized population of Boston which is counter-intuitive [Segata, Current Biology, 2015]. The stratified profiles of HUMAnN2 are not helpful in this case.

Our current design of Carnelian does not support functional profiles stratified by species. However, the level of extra information HUMAnN2 provides can be easily obtained by running off-the-shelf taxonomic profilers such as OPAL or Kraken1/2 on the input reads and then quantifying the taxonomic contribution to microbial functions using the genomic content information of known species. Of course, this approach has all the same problems HUMAnN2 has with respect to most reads not corresponding to any known genome. Given that practitioners in microbiology widely use aggregate functional profiles (without stratification) while investigating interpersonal differences in communities depending on disease conditions,

geography, lifestyles, etc. [Qin et al., Nature 2012; Karlsson et al., Nature 2013; Knoll et al., American Journal of Physiology-Gastrointestinal and Liver Physiology, 2016; Pasolli et al., Cell, 2019, etc.], we felt that aggregate profile construction is itself an important problem to tackle.

Cross-sample comparability

The authors claimed that one major advantage of Carnelian is that results can be compared cross samples. They detailed their design in page 3 lines 49-60. The main change over existing tools is that they normalized hit numbers by "effective protein length", the equation of which is provided in page 7 line 11. However, I do not see the justification why this metric is superior to no normalization. I can imagine that this equation is to provide correction given the short nature of Illumina sequencing reads versus the long nature of whole proteins in the reference database. But the authors need to provide explanation, as well as evidence in both theory and benchmark tests to demonstrate why it is better and how it represents a significant innovation.

We apologize for not having included our thought processes in designing the normalization. To emphasize why this normalization step is important, we newly discuss the intuition behind it and provide additional experiments in the updated Supplementary Note X2 in Additional File 1. We also include the discussion here for ease of review.

Let's look at the intuition behind our normalization step. The input to Carnelian's pipeline is short metagenomic reads from a metagenomic sample and our goal is to determine relative abundances of functional terms (in this case Enzyme Commission (EC) labels) present in the sample. The default assumption is that each coding read represents some gene in part in the microbial sample. We translate these reads to the best possible open reading frames and then bin them into ECs according to some learned representations of proteins from a reference database. All else being equal, the more abundant proteins from an EC bin in the microbial sample is, the more reads from them are likely to be sequenced. Therefore, read counts can be used as a proxy for EC abundance in the sample --- used by common functional annotation tools (e.g. mi-faser). However, in practice "all else" are never equal. These counts need to be made comparable across proteins, samples, and experiments to enable meaningful comparative analysis.

Let's consider the following scenarios.

Scenario 1: Suppose, a microbial sample has only two proteins (from two different ECs) in equal proportion. These protein sequences have different lengths. If we sequence the sample, there is a high possibility that we will see more reads from the longer protein (thus more reads from the corresponding EC). If we take raw read counts as a proxy for relative EC abundance, we will mistakenly assume that the EC with the longer protein is more abundant. This is the reason we need to normalize the read counts in an EC bin by the effective protein length (the positions in the protein sequence to which a read can actually map) of that bin. This value is often known as RPK when the length is measured in kilobases (used by different methods such as HUMAnN2).

Scenario 2: Suppose, we have reads from two experiments with different sequencing depths --- one experiment has 10x more reads than the other. If we want to compare the relative abundance

of the same EC across experiments, just normalizing by effective protein length in the corresponding EC bin will not change anything. The higher the total number of reads, the higher read count we should expect for any given EC. For relative abundances to be comparable across experiments they need to be on the same scale.

Scenario 3: Suppose we have two microbial samples each with two types of proteins (from two different ECs). Sample 1 has red and yellow proteins and sample 2 has red and green proteins. The lengths of red, yellow, and green proteins are 10, 50, and 250 units respectively. Let's say, we observe 300 reads from both samples and we want to compare the abundance of red proteins across samples. If we observe 50 reads from the red protein in both the sample, the RPK values for red protein will be the same across samples. We observe 250 reads from the yellow protein in sample 1 which means a relative abundance of red protein is much less compared to yellow protein here (RPK for red protein = 1×10^3 vs. RPK for yellow protein = 25×10^3). In sample 2, we observe 250 reads from the green protein which means both red and protein have the same relative abundance (RPK for both proteins is 1×10^3). This means sample 2 has a higher abundance of red protein which we will not be able to tell if we only compare the RPK values. Clearly, the RPK values of other proteins in the sample have an effect on the relative abundance of a protein in question. If we normalize by the sum of all the RPK values in the sample, then we can see the desired difference (normalized RPK values of red protein in sample 1 and 2 are $1/26$ and $1/2$ respectively).

Carnelian takes the above scenarios into account and normalizes read counts by effective protein length in the EC bin and a per million scaling factor which incorporates the sum of all RPK values in the sample. This normalization ensures that the relative abundances of the EC bins in every sample effectively sums up to the same number making them directly comparable across samples.

Experiment:

We newly perform an experiment to show how well this normalization works in practice. We randomly selected an individual from our Bostonian cohort. The original read data set contained ~9M paired-end reads of length 150 base pairs. We created another read data set by performing 20x downsampling such that the new subsampled data set has ~450k reads. Ideally, the relative abundance of all enzyme commission labels (ECs) should be the same in these two samples and we should observe a log fold-change (logFC) of zero (0) for all of them. We used raw read counts (used by mi-faser), RPK measure (used by HUMAnN2), and our effective read counts (TPM measure) as proxies for relative abundance and measured the logFC value for all the ECs in each case. While nearly every EC appears variable between the original and the subsampled data set in terms of raw read counts and RPKs, only Carnelian's effective counts show the expected behavior.

	Raw count	RPK	Effective Count
Mean logFC	-1.1251	-1.1251	0.0094
Stddev logFC	0.4659	0.4659	0.1918

Some examples:

EC	Raw count		RPK		Effective Count	
	Original	Subsampled	Original	Subsampled	Original	Subsampled
3.6.3.19	1090	56	2939.99	151.05	2072.43	2120.91
1.2.1.11	876	45	2442.38	125.46	1721.66	1761.72
5.2.1.8	935	48	3231.71	165.91	2278.07	2329.58
3.6.3.25	2689	138	8280.22	424.94	5836.82	5966.87
4.1.1.87	78	4	187.95	9.64	132.49	135.34
2.7.7.24	3096	153	10927.06	540.00	7702.60	7582.46
4.2.1.24	466	23	1401.50	69.17	987.95	971.30
6.1.1.19	223	11	389.37	19.21	274.47	269.69
4.2.1.8	981	48	2468.74	120.79	1740.24	1696.15
2.5.1.47	3148	154	10092.68	4937.34	7114.44	6932.81

Actually, when I first see the claim of cross-sample comparability in the abstract, I was expecting to see things like ranking, log-ratio transform, etc, which have been widely experimented by recent researchers. The current treatment in Carnelian is quite simple and I am not convinced that it represents a major innovation.

We agree that there is still much work to be done for cross-sample comparability, and many treatments of it in the literature are much more mathematically sophisticated than our conceptually simple normalization step. However, existing standard functional profiling tools do not make use of *any* such approach. While we do not know why this is so, part of the reason may be that these transformation-based approaches lack interpretability. Thus, although a read-count normalization may not be a major innovation, it is a positive advancement that works (as shown above) and is straight-forward to explain to practitioners and users of our software.

Multiple assignment

Page 11 line 23: "We excluded any protein that had computationally inferred functional labels (e.g. by homology), an incomplete EC label, or multiple EC annotations".

It is very common for a protein to possess multiple functions--how does Carnelian deal with this situation? If each protein is assigned to only one EC, I would be very unconvinced by any claims of the goodness of results.

We agree with the reviewer that many proteins can have multiple functions. However, these proteins primarily act as enzymes and the **secondary functions are mainly non-enzymatic**. Therefore, it is a safe assumption to make that proteins will have only one EC label because we are focusing on only the metabolic/enzymatic functions of proteins for this study. We include this reasoning in the database curation subsection of the updated Methods section.

ORF finding

The feature "probabilistic ORF finding" was emphasized multiple times including the abstract. But wait, is it just FragGeneScan, a simple tool published 10 years ago and already widely-used? I am not saying that the authors cannot describe FragGeneScan as "probabilistic ORF finding". I just think it is not worth stating it in such way that it sounds like an innovation made by the authors.

Given the reviewer's criticisms, we have rephrased our references to FragGeneScan and removed it as a key selling point of Carnelian, as it is but one of many steps in our pipeline.

Reliance on EC terms

The basic units of the Carnelian reference database are enzyme commission (EC) terms. EC terms are commonly used and are a nice feature in many functional annotation tools. However, not all proteins are enzymes. Reporting EC terms only is a limitation in many use cases. Other protein function catalogs are usually more comprehensive. For example, gene ontology (GO) terms have three top-level categories: "cellular component", "molecular function", and "biological process".

The authors noticed this and described Carnelian as "focused on metabolic functionality", "as opposed to using typical protein database..." (page 2 line 47). This is accurate, however I do not see why this is an advantage instead of a limitation ("(Carnelian) is better suited for..."page 2 line 43). This needs to be justified.

Our focus on metabolic functionality is both an advantage and a limitation. As the reviewer correctly points out, other annotation tools cover a wider array of functionality. However, the purpose of our study is to provide a method that is most suited to finding microbial functional changes related to human health and disease. While proteins can have many non-enzymatic functions, such as housekeeping functions like transcription, translation, structural components, information processing, etc., their metabolic functionality is the most relevant one when it comes to host health [Rath et al., Microbiome, 2018] -- which is why we chose to focus solely on it. The housekeeping functions of microbial proteins are essential for microbial existence and may not provide much useful information while comparing healthy and diseased microbiomes.

While it is informative to look at all the functions when characterizing a particular species, in Carnelian, we provide an alternative that does not spend compute time and resources on matching reads to housekeeping functions. For studies that care about those functions, Carnelian

should not be used with the curated EC database which is focused on metabolic functionality; however, when we are aiming to compare hundreds of microbiomes and are primarily interested in changes in metabolic functionality and host health, Carnelian's focus (and concomitant increase in performance on this task) is an advantage.

We should also note that the choice of database is study-specific; Carnelian is not tied to the EC database. It can be trained and used with any database including the entire eggNOG, KEGG, or UniRef database. We now make a pre-trained model of 1,785,722 UniProt proteins from 3,285 COG functional categories available through Carnelian's website (See database curation subsection in the updated Methods section). These proteins and their COG annotations are publicly available from the eggNOG database (<http://eggnogdb.embl.de>). This database is not focused on metabolic function and can be used for general-purpose functional profiling that goes beyond metabolism. Since KEGG is a proprietary database, we do not provide a model for KO functional groups. Users who have access to the KEGG database can easily train a model following the instructions on our website.

Benchmark

Precision

The authors did present precisions, F1 scores (Note X3) and AUCs (Fig. 2). Why didn't the authors also emphasize on them in the main text? One possible reason I can imagine is that Carnelian's precision is not as high as other tools (especially HUMAnN2, e.g., Fig. FX4, Table TX8-9). Of course, the difference isn't dramatic, but meanwhile the increase in sensitivity isn't dramatic from those tables & figures either.

This result is (partially) in my expectation, since HUMAnN2 is based on amino acid sequence alignment (and a database of marker genes instead of whole genomes, but here the authors used a uniform database), being accurate at the cost of lower sensitivity, whereas Kraken is known suffering higher false positive assignment due to the nature of k-mer matching. But it is bit surprising that Kraken2 performed worse in both precision and recall according to Fig. FX4.

We want to emphasize that our primary goal in this study is to enable functional comparison across diverse data sets, which we believe our experiments in the main paper validate. Although part of this necessarily involves benchmarking to ensure that our profiling results on single data sets are reasonable, we specifically do not cast Carnelian as a general-purpose replacement for existing tools---indeed, this is also why we focused entirely on only enzymatic actions, rather than labeling all types of protein activity.

It is true that the improvements to F1-score are not dramatic, and that we are clearly trading off on precision vs. sensitivity. However, we would like to emphasize that benchmarking the task of functional inference itself is hard due to the lack of gold standard functional labels. We agree with the reviewer that alignment-based tools like HUMAnN2 and mi-faser (both wrappers of DIAMOND) are highly precise. As the reviewer pointed out, Kraken2's use of minimizers and exact k-mer matching makes it prone to false positives which explains the reduced precision. We believe its reduced recall stems from the fact that it performs an all-six-ORF translation using the

standard codon table before performing classification. Since prokaryotes have different codon usage bias, this step possibly introduces error. Since the new Kraken2 algorithm is not published yet, we are not aware of the details of it.

Database

For a fair comparison, the authors used the same reference database to benchmark all four tools. This practice is correct and should be encouraged. However, therefore the analysis shown here was actually benchmarking sequence aligners, not functional profilers.

We thank the reviewer for agreeing with us on the importance of benchmarking all four tools against the same reference database. However, we disagree that the analysis shown here was only benchmarking sequence aligners. While most existing tools (including the three we compared against) do have a sequence-alignment component, we are still interested at the end of the day only in their functional profiling results (Experiments described in Supplementary Note X5.1). Each tool has its own set of parameters, similarity thresholds, and post-processing steps for assigning functional labels to reads. Given that Carnelian does not have a sequence-alignment intermediate step, it is not possible to directly benchmark sequence alignments.

Some dedicated functional profilers, such as HUMAnN2, has its official database or protocol for making database (ChocoPhlAn + UniRef + MetaCyc). Hooking HUMAnN2 to the Carnelian database and showing it is less sensitive than Carnelian seems to be unfair.

Our decision to use the same database was based on our desire for a “fair” comparison. Analyzing against custom databases is of particular importance when working with less well-studied populations, so we think that is a relevant metric.

However, given the reviewer’s misgivings, we do as they suggest and newly ran out-of-the-box HUMAnN2 with default databases on the microbiomes of all the non-industrialized Baka individuals (our novel in-house data set). Each sample has ~7M paired-end reads of 150 bp length. On average HUMAnN2 could annotate only ~10% reads per Baka sample despite using the entire ChocoPhlAn and UniRef database. On average, HUMAnN2 detected ~30 species and 996 ECs per sample. The average Shannon diversity index per sample was 5.58. We also ran out-of-the-box HUMAnN2 on the microbiomes of 20 randomly selected industrialized Bostonian individuals (our novel in-house data set) for comparison. Since industrialized microbiomes are well characterized, here HUMAnN2 can annotate more reads (~40-50% per sample) and detect more ECs per sample (Shannon diversity index 5.95). But, the literature suggests we should see more diversity in non-industrialized populations compared to industrialized ones [Segata, Current Biology, 2015] which we observe with our curated EC database. We now discuss these findings in the third subsection of the Results section. The details of this experiment are provided in Supplementary Note X6 of Additional File 1.

Additionally, as we mentioned in our response to Reviewer 1’s comment on taxonomic assignment, we also ran full HUMAnN2 pipeline with default databases on our case-control Parkinson's disease data set and compared HUMAnN2-generated MetaCyc pathway profiles between cases and controls. It did not identify any pathways related to tryptophan metabolism

which is a clinically established hallmark of Parkinson's and was identified by both Carnelian and the assembly-based gene catalog approach of the original study [Bedarf et al., 2017]. The differentially abundant pathways identified by HUMAnN2 were largely related to purine and pyrimidine metabolism which is non-specific to Parkinsonism. We now discuss this in the fourth subsection of the updated Results section and provide the list of HUMAnN2-detected MetaCyc pathways in Supplementary Table T46 of Additional File 6.

And why don't the authors directly benchmark Carnelian against protein sequence aligners, such as BLASTP, DIAMOND, HMMER, etc.?

We did not perform benchmarking separately against protein sequence aligners (e.g. Blastp, psi-blast, RapSearch2, hmmer, etc.), because HUMAnN2 and mi-faser already use the state-of-the-art aligner DIAMOND as part of their functional labeling pipeline; DIAMOND is orders of magnitude faster than other protein sequence aligners at comparable precision and sensitivity [Buchfink et al., Nature Methods, 2014]. Moreover, mi-faser demonstrates better sensitivity and precision than blastp, psi-blast, and hmmer [Zhu et al., NAR, 2018], so we felt comparing against HUMAnN2 and mi-faser should suffice.

Kraken

I do not know Kraken1/2 was designed and promoted as a functional profiler. It is a taxonomic profiler. The authors need to explain why they included Kraken in the benchmarks.

And, Kraken features a heuristic based on a taxonomic hierarchy. Does the Kraken database built by the authors also have any sort of hierarchy? If not this is another source of unfairness.

While Kraken1 was designed only as a taxonomic classifier, Kraken2 includes a new protein search feature (<https://ccb.jhu.edu/software/kraken2/>), which can readily be used for functional profiling. We chose it for comparison as it is a comparable state-of-the-art tool that uses a compositional k-mer based approach. We built the protein database exactly as described on their website and ran on data sets following the Kraken2 manual. Commands were provided in the original Supplementary Note X6 (Supplementary Note X9 in the revised manuscript).

During the protein search, we believe Kraken2 does not use the taxonomic hierarchy, it performs a six-frame translated search of the query sequences against the protein database. Therefore, using our flat database for Kraken2 protein search is not a source of unfairness.

Precise mode

I noticed that Carnelian has a "precise" mode (Note X4). I wonder how precision / sensitivity would be in this mode?

Yes, Carnelian does have a precise mode that uses one-against-all logistic regressors to perform functional binning. We make this option available in case users want to look at the probabilities assigned to the functional labels associated with a read. However, all the results presented in the main manuscript were generated using Carnelian's default mode which uses an ensemble of SVM classifiers and performs much better in practice than logistic regression. Therefore,

discussions related to the performance of Carnelian in precise mode were not included in the manuscript. We now make the performance of Carnelian in precise mode available through Carnelian's website (<http://carnelian.csail.mit.edu>).

Test dataset

The authors created a synthetic community for benchmarking, as shown in Table TX5. However, a microbiologist will quickly realize that it does not look like a realistic human gut microbial community. To justify this design, the authors stated that they are the 20 most abundant bacterial species in HMP stool samples. I am not clear how this information was drawn. But in Fig. 3a of Huttenhower et al. Nature, 2012 (which the authors cited) indicated otherwise.

I suggest that the Carnelian team have a microbiologist reviewing their experiments, if not already.

The synthetic metagenome was created following a similar strategy to the paper introducing HUMAnN2 [Franzosa et al., 2018] (Reference 8 in the original Supplementary Notes) which we mentioned in the text (Supplementary Note X5.1 in Additional File 1), but it might have been missed. The 20 most abundant species used in the design were originally identified by the Huttenhower et al. Nature, 2012 study. We have Eric Alm and his lab carefully reviewing our experiments and incorporating Carnelian into their pipelines because other methods fail to compare function across diverse populations.

Statistical test

To test the differential clustering of metagenomic functional profiles by certain metadata categories (e.g., industrialized and non-industrialized), the authors relied on the PERMANOVA test, which is a good choice for this question. However, the interpretation of PERMANOVA result is problematic. The authors report p-values alone. But what is more important for PERMANOVA is the pseudo-F statistic, which indicates the ratio of inter-category variance over intra-category variance; and the R-squared value, which indicates the proportion of variance explained by this metadata category. Both are available from the "adonis" function which the authors used, and should be reported. In contrast, the p-value of a PERMANOVA test is not a good indicator of group separation significance, and is almost always small given the sample size is large. (Strangely, the p-values shown in Fig. 3b seem to be very large.)

The pseudo-F statistics from the adonis function are now reported in conjunction with the associated p-values in our modified Figure 3. However, there is plenty of literature which argues against using R-squared value as a goodness-of-fit measure or as a measure of how one variable explains another [Anderson and Shanteau, 1977; Birnbaum, 1973; Berk, 2004; <http://www.stat.cmu.edu/~cshalizi/mreg/15/lectures/10/lecture-10.pdf>; <https://data.library.virginia.edu/is-r-squared-useless/>]. Therefore, we deliberately exclude R-squared values from the discussion.

Introduction

The authors reviewed the previous and current trends on the topic of shotgun metagenomic data analysis and functional profiling. I noted that many representative studies and tools have been mentioned, which I appreciate. However, I found the overall logic misleading.

Abstract and page 2 line 9: "these attempts have generally been based on targeted 16S rRNA sequencing data..." Come on. This is 2019. A large number of studies already adopted shotgun data, and even multi-omics data.

We were referring specifically to the large meta-analyses cited, which were based on prior studies on primarily 16S rRNA data. However, we have removed the reference to 16S rRNA data in light of much of new data being shotgun sequencing moving forward. Instead, we specifically point out in the text that those meta-analyses were unable to find either shared taxonomic or functional dysbiosis, leading to the aims of our study (second paragraph of the updated Background section).

Another reason suggested by the authors is that existing tools "are limited in their ability to directly compare microbial metabolic function across samples and studies." I have detailed my critique above.

Page 1 line 59: "though quite related to functional profiling..." Bit distracting. What is the main point?

Upon further reflection, we have decided to simply remove that clause. We thank the reviewer for bringing it to our attention.

Page 2 line 13: "...lack of sensitivity of existing functional annotation tools, inconsistencies in annotations... across... databases, and lack of comparability of existing... statistics". These critiques are strong, and need to be backed by evidence. Despite the authors showed that in very specific experiments Carnelian may have higher sensitivity, but more are expected in order to make these claims.

The reviewer is correct that our claims about other methods were too broad. We have rewritten the introduction and now refer more specifically to the deficiencies directly justified by our experimentation; i.e. we now no longer use sensitivity as a primary benchmarking method, but we do stand by our claim of the lack of comparability. That is the very point of (1) our experiments showing that other tools were unable to detect concordant functional shifts due to the same disease across different geographical cohorts and (2) our experimentation using standard pipelines without our renormalization, where comparability is not even present in the full vs subsampled data sets.

Biology

Page 8 line 60: "nonindustrialized microbiomes showed enrichment of reads in antibiotic resistance ECs and pathways (e.g. beta-lactamase...": Why would less industrialized communities have higher abundance of antibiotic resistance genes? This is not intuitive.

Higher abundance of antibiotic resistance genes in non-industrialized communities is a well known phenomena among the microbiology community and has been reported in several studies [Rampelli et al., 2015; Pehrsson et al., 2016; Baron et al., 2018]. There could be a multitude of

potential reasons for it, including inappropriate use of antibiotic drugs, poor drug quality, close contact with soil, uncontrolled use of antibiotics in livestock, close contact with animals, and low sanitation. We have now added appropriate references to the discussion (fifth paragraph in the updated Discussion section).

Terminology

The authors described Carnelian as a "compositional" tool, and "compositional" means "k-mer based" (page 2 line 39). I find this term ambiguous. First, in traditional genomics, the term "compositional" also describes GC-content, tetramer frequencies, codon usage signature, etc. Second, "compositional" can be easily confused with the statistical term "compositional" (meaning adds up to one) which has been frequently used to characterize microbiome data. Therefore, I would suggest the authors be more specific. Why not just using "k-mer based"?

We thank the reviewer for this suggestion, and now refer to Carnelian as a "gapped k-mer" tool in most parts of the main paper. One exception is that we continue referring to Carnelian as "compositional" in our discussion of the machine learning classification components (though we still add in "gapped k-mer" as a parenthetical). We made this choice because the lineage of k-mer classifiers Carnelian is based on [Vervier, et al., 2016; Luo, et al., 2018] refer to their SVM methods as "compositional." Hopefully, this reduces confusion in the traditional genomics community while still preserving continuity with terminology in the bio-ML literature.

Formatting

The main text PDF is gigantic. The figures seem to be formatted in an unfavorable way, which slowed down my computer when I was trying to view them. Please fix. Line numbers are not incremental across pages and are not aligned to each line. Hard to track. Please reformat.

Unfortunately, the final PDF is generated automatically by Genome Biology's submission system from our source files and we have little control over it. The editor indicated in our decision letter that this is not a concern.

There are two Table TX6's. Indexing problem?

We thank the reviewer for pointing that out. We have now corrected the indexing problem in the updated Supplementary Notes file.

Final Words

Based on the reasons detailed above, I do not agree that "(Carnelian) newly enables using whole metagenome sequencing for meaningful large-scale comparative functional metagenomic studies across diverse populations.". I think this work needs to be substantially reworked before it can be considered for publication.

We have carefully addressed the comments of the reviewer to the best of our abilities. While we agree that the original framing and metrics (especially surrounding the use of sensitivity, as the

reviewer rightly points out) were problematic, we hope that the substantial changes we have made, the additional experiments performed, and the new explanations provided will be convincing to the reviewers and editors.

Reviewer #2: In this paper, the author demonstrated an effective method for metabolic functional profiling of metagenomic studies. The Carnelian is a useful tool for meaningful large-scale comparative functional metagenomic studies across diverse populations.

We appreciate the positive assessment.

Some specific comments and suggestions are as follows:

1. Section "results", Second paragraph, "Thus, we constructed our gold standard reference database with curated prokaryotic proteins that have verified unique and complete EC labels which provide a direct mapping to KEGG metabolic pathways for our later analyses.". How big is your database and how many sequences it contains? What is the difference between this database and the known database, and how many proteins are newly added? Please be more precise in the results.

The database we used for this study consists of 7,884 prokaryotic proteins with 2,010 unique EC numbers (originally mentioned in the Database Curation subsection of Methods section). We now explicitly mention this in the Results section as well (second paragraph of the beginning of the Results section). A previously-available curated EC database from mi-faser [Zhu et al., 2018] contained 2,810 prokaryotic proteins with 1,257 unique EC numbers.

2.The ORF detection must depend on the parameters of the program, what were the parameters used by the authors, so how robust is this conclusion?

FragGeneScan comes with pre-trained models (parameters for HMM) for different settings and requires the user to tell only whether the input file has short reads or complete genomes and how much is the sequencing error rate. By default, Carnelian uses the short reads version of FragGeneScan. Since the average substitution error rate for illumina sequencing is ~ 0.1%, we used the 'complete' option with FragGeneScan which assumes 0% error rate and is sequencing technology agnostic. We now make this point clearer in Methods (third paragraph of the first subsection). By design, FragGeneScan does not use statistical parameters such as the length distribution of genes and reads while calculating the probability of gene regions and non-coding regions. These probabilities are calculated solely based on the composition of input reads. This makes its predictions robust for a wide range of read lengths [Rho et al., NAR, 2010].

3. Section "Carnelian reveals novel and shared functional dysbiosis across disease studies", paragraph three, "At the pathway level, we found the PD gut to have lower read abundance in several carbohydrate metabolism pathways (BH-corrected Wilcoxon rank sum test p-value <

0.05 and absolute log fold change > 0.11)", what are the specific carbohydrate metabolism pathways? A supplementary table with those pathways may be interesting for the reader.

We thank the reviewer for this suggestion. The list of differentially abundant pathways is available in the Supplementary Table T42 (Additional File 6) of the updated manuscript (Supplementary Table T5 of Additional File 2 in the original manuscript). For more clarity, we have newly added the broader category of pathways to the table in the updated supplementary which will make it easier to identify the carbohydrate metabolism pathways among them.

4. Section "Carnelian reveals novel and shared functional dysbiosis across disease studies", paragraph six, "Carnelian also finds lower abundance of several EC terms including 1.2.1.19, 3.5.1.16, 4.1.1.18, and 5.3.3.10 which play crucial roles in the metabolism of several essential amino acids including arginine, proline, lysine, tyrosine, etc.". the reader may not be expected to know by heart the meaning of all digital terms notations, explanations need to be provided throughout.

We thank the reviewer for this suggestion. We have now provided the names and roles of the enzymes in addition to the EC numbers for additional clarity in the Results section of the updated manuscript.

5. Is the inconsistency of the metabolic pathways detected in Carnelian, mi-faser, HUMAnN2 and Kraken2 mainly due to the difference in the database on which each software depends, or the difference caused by the algorithm of the software itself?

We apologize for not being clear. All four tools were trained and run on the same reference database for all the data sets to ensure that the differences in results are purely due to the performances of the algorithms. While we had originally put this detail in the Methods, we now make this point much clearer in the main manuscript at the start of the Results section (end of first Results subsection).

Second round of review

Reviewer 1

In this revised manuscript and response letter, the authors made rigorous modifications and reasonings to address the points raised by me and the other reviewer. The new materials are very comprehensive and relevant. Despite my previous comments were largely critical, the authors showed respect and patience, which I appreciate. I think the quality of this manuscript has been substantially improved.

Overall, I see the value of this work (especially in the revised form). First, I think it is worth adding a k-mer based functional profiler to the shelf of microbiologists. I agree with the authors' statement: "Importantly, they (k-mer-based profilers) can be trained to directly classify WMS reads by function, even when the read itself comes from a protein that is not in existing databases." This capability enables interesting potential for assessing the functionality of under-characterized microbes.

Second, the revised work highlights cross-sample comparability, which the authors demonstrated using several datasets. While precisely defining the functional role of individual ORFs is difficult, exploring the overall trend without local accuracy is sometimes acceptable and demanded by the community.

Philosophic question: The function of a protein is usually determined by its 3D structure, the highest rank in a cascade of information: nucleotide => codon => amino acid => motif => domain => subunit etc. Previous researchers have explored structure-based alignment with success to certain extent. The authors of Carnelian chose the reverse direction, which is interesting (though not thoroughly explored in this article).

However, I still have multiple concerns of the revised work and the responses to my questions—some of which are substantial—as detailed below.

Novelty.

I am not convinced that Carnelian represents a substantial innovation over current methodology. Cross-sample comparison has been done by many, using for example HUMAnN2 and other tools. Strong biological signals have been demonstrated in previous publications. The Carnelian team did a good job in demonstrating the advantage of their program in this task on selected datasets. Yet, I see this advantage as incremental, not game-changing. Second, I don't think the exploitation of k-mer matching in protein alignment is unprecedented. The authors already noted Kraken2. And I want to point the authors to a recently published protein sequence searcher: MMSeqs2 (Steinberger & Söding, 2017, Nat Biotech), which is gaining popularity in the field. The program uses a multiple layer engine, with the first of which being similar-k-mer matching (though not gapped), followed by ungapped and gapped sequence alignment.

Comparison with other tools. The authors made efforts to justify the choice of HUMAnN2 (native and uniform databases), Kraken2 and mi-faster to benchmark against Carnelian. They make some sense and I appreciate that. However I am not convinced that the current strategy alone is most meaningful. These tools being complex functional profilers (with the exception of Kraken2) naturally fail to satisfy the necessity of variable controlling in a scientific experiment design. As the authors admitted, this is not an "apple-to-apple" comparison. I see the marketing value but I am afraid that the scientific rigor is impaired.

In my opinion, the most relevant tools to include in benchmarking are the gapped k-mer binner (backend of Carnelian), DIAMOND (sequence alignment), Kraken2 (k-mer matching), HMMER (profile HMM), and MMSeqs2 (combined search engine). One shouldn't simply assume "DIAMOND is the state-of-the-

art” and skip this basic step.

With this experiment completed, I think the comparison of complex functional profilers is more reasonable, as it informs the users what they can get from each tool out-of-the-box. I appreciate the authors’ addition of HUMAnN2 out-of-the-box mode in the comparison. I am also curious: what about PICRUSt (Langille et al., 2013), which predicts function simply based on taxonomy? We are aware of its limitation in accuracy due to say horizontal gene transfer (as the Carnelian authors also mentioned). But regarding the cross-sample comparison task, I am curious how it will perform.

Sensitivity and spurious labels: The authors addressed my concern with de-emphasization of sensitivity (though left some, for example page 4 line 34) and discussions on the issue of spurious labels. I appreciate this but my concern remains with the revised form. The 2nd last paragraph of page 9 have multiple “we believe that...” I think these are over-claims that are misleading, especially when the lower precision of Carnelian is already evident. More problematic is page 3 line 21: “classify patients vs control ... Carnelian classifications are not spurious”. I do not see the logic here. I understand that experimental validation is difficult, but it is necessary if one wants claim against spurious labels. Therefore I would suggest the authors focus on cross-sample comparability but don’t over claim accuracy.

Stratification: The authors presented intense reasoning for not performing taxonomic binning of functional units in Carnelian. I partially agree. What I agree with is that stratifying functional units by taxonomy is a difficult problem, and HUMAnN2’s solution is far from being perfect. HUMAnN2 has its inherited limitation by adopting MetaPhlAn2, which is marker gene based, with high precision but low sensitivity comparing to alternatives (Ye et al., 2019, Cell). I also agree that stratification can be assessed by pairing Carnelian with a taxonomic profiler.

Though, admittedly, Carnelian’s without taxonomic stratification (while HUMAnN2 has) is a limitation for users, many of whom do have this demand.

The authors were right that a large number of studies have focused on aggregate functional profiles. But another large number of studies involving shotgun metagenomics have explored genome-level variation, using read-based or assembly-based techniques, the latter category of which has increasingly becoming practicable and standardized. I would thank the authors for pointing me to Pasolli et al., 2019, Cell.

However, I am afraid that their interpretation of this important work is incorrect. Pasolli et al.’s starting materials were species-level genome bins (SGBs), that is, they were already assembled and binned (a process the Carnelian authors deemed difficult), thus stratification was not necessary. Actually, Pasolli et al. Intensively discussed the characteristics and variation at genome level, including functional profile, as evident in their Tables S6 and S7.

Effective protein length: I appreciate the authors’ additional work in elucidating and validating the effectiveness of this metric. The explanation is very clear and helpful. This notion is overall fine to me. I agree that comparing to more complicated math, the simplicity and interpretability of effective protein length is an attraction, especially in the field of functional profiling where advanced maths have not been thoroughly explored.

However, I hope the authors note this question in the manuscript. It is a crucially important, yet not well-answered question. Carnelian’s treatment, which considers per-sample relative abundance of functional proteins, is at the edge of the compositionality trap, and could also be confusing to some people (why functional units add up to one?). I hope the users are informed, and more ideally, have the option of keeping intermediate raw hit counts, on which users can experiment various maths.

Synthetic metagenome. The protocol for designing the human gut microbiome sounds overall fine to me. But the authors missed two things: 1) "top 20 most abundant species" does not mean a typical gut microbiome is mainly composed of these species together. 2) Sorting species name by alphabetical order and applying a simple geometric distribution may not be ideal.

The species list in Table TX6 are indeed common human gut associates. What I questioned was the distribution of relative abundances. Alistipes can't be this high (60+%). Prevotella, Ruminococcus and Faecalibacterium are typically higher than 0.0x%. There should be several percentages of Actinobacteria and Proteobacteria. Actually, the last of which is among the main sources of profiling challenge due to their genomic flexibility and pathogenic potentials. (There are also several dozen percentages of dark matters, but let's not complicate things here.)

I would suggest the authors consult Eric Alm and his crew again with this specific question. Multiple assignments and reliance on EC terms. I accept the authors' reasoning, considering that Carnelian mainly predicts enzymes, and that multiple existing tools and protocols also don't resolve this issue well, for example they only take the hit with best alignment score, or simply the first hit come across in database search. I also appreciate that the authors generated and provided the COG model.

PERMANOVA. I do not agree with the authors' defense against reporting R2 value in a PERMANOVA test. I understand that some people don't like R2, just as some people (likely many more) don't like p-value (for example see Amrhein et al., 2019, Nature: Scientists rise up against statistical significance). Nevertheless, I don't know any decent journal would not demand reporting p-value when available. Admittedly, R2 is by far the most widely used and basic metric for assessing the effect size of variables. Therefore, I would still recommend that the authors report R2 values, together with pseudo-F and p-values.

Unpublished datasets. My understanding is that the Bostonian and Baka datasets have not been published, and the authors have no intention to make them publicly accessible. This blocks the chance for readers to replicate the bioinformatics described in this article. I am not sure if this practice complies with the Genome Biology policy. But I will leave this question to the editor.

Reviewer 2

Minor points:

There are still a few typos and grammar errors, say:

1. Page 3, line 41. delete "in order".

2. Page 3, line 46-49. this part needs to be rewritten.

3. Page 5, line 43-44. Change the "WMS data comes from a mixture of many different organisms, and can encode 100x more unique genes than are present in just the human genome" to "WMS data comes from a mixture of organisms, and can encode 100x more unique genes than those present in just the human genome".

4. Page 8, line 49 change the "For completeness we also performed the classification experiments choosing EC labels from the entire data set" to "For completeness, we also performed the classification experiments choosing EC labels from the entire data set".

General Comments to Reviewers (in blue):

We wholeheartedly thank both reviewers for their efforts at improving our manuscript. In summary and as directed by the Editor, we have addressed the following concerns:

- *Comparison with similar methods, particularly MMSeqs2.* We have newly provided a comparison of performance between Carnelian's gapped k-mer based binning method and widely-used protein search tools, DIAMOND, PHMMER, and MMSeqs2.
- *The distribution of relative abundances for the simulated data.* In consult with the Alm lab, we have constructed a synthetic microbiome dataset with similar relative abundances to the human gut and favorably compared Carnelian to other methods on the dataset.
- *Better highlight cross-sample comparability.* We have highlighted our unique contribution to cross-sample and cross-population comparability in the updated manuscript.
- *Threshold choice.* We are not exactly sure what is meant by "threshold choice" because we don't choose any thresholds for the methods. Perhaps the Editor's comment is related to, "variable controlling in a scientific experiment design", which we address below.
- *Data release:* The data from the Bostonian populations are now available from NCBI SRA and the sample accession numbers are provided in the updated manuscript as well as the Carnelian website. Since the Baka are an indigenous people, the data from them is sensitive and required to be released only through an access controlled repository. The Broad Institute is preparing to release the entire Baka dataset through dbGaP shortly; we will work out a solution with the editor about what to do in the interim.

Point-by-point Responses to Reviewers (in blue):

Reviewer #2: Minor points:

There are still a few typos and grammar errors, say:

>> We thank the reviewer for his/her careful reading of and positive comments on our manuscript. We have addressed the comments and highlighted the changes in the revised manuscript.

1. Page 3, line 41. delete "in order".

>> It has been deleted in the revised manuscript.

2. Page 3, line 46-49. this part needs to be rewritten.

>> Thanks for pointing this out. We have re-written this part in the revised manuscript.

3. Page 5, line 43-44. Change the "WMS data comes from a mixture of many different organisms, and can encode 100x more unique genes than are present in just the human genome" to "WMS data comes from a mixture of organisms, and can encode 100x more unique genes than those present in just the human genome".

>> Thanks for the suggestion. It has been changed in the revised manuscript.

4. Page 8, line 49 change the "For completeness we also performed the classification experiments choosing EC labels from the entire data set" to "For completeness, we also performed the classification experiments choosing EC labels from the entire data set".

>> It has been changed in the revised manuscript.

Reviewer #1: In this revised manuscript and response letter, the authors made rigorous modifications and reasonings to address the points raised by me and the other reviewer. The new materials are very comprehensive and relevant. Despite my previous comments were largely critical, the authors showed respect and patience, which I appreciate. I think the quality of this manuscript has been substantially improved.

Overall, I see the value of this work (especially in the revised form). First, I think it is worth adding a k-mer based functional profiler to the shelf of microbiologists. I agree with the authors' statement: "Importantly, they (k-mer-based profilers) can be trained to directly classify WMS reads by function, even when the read itself comes from a protein that is not in existing databases." This capability enables interesting potential for assessing the functionality of under-characterized microbes.

Second, the revised work highlights cross-sample comparability, which the authors demonstrated using several datasets. While precisely defining the functional role of individual ORFs is difficult, exploring the overall trend without local accuracy is sometimes acceptable and demanded by the community.

>> We gratefully thank the reviewer for recognizing the potential of our work after revision.

Philosophic question: The function of a protein is usually determined by its 3D structure, the highest rank in a cascade of information: nucleotide => codon => amino acid => motif => domain => subunit etc. Previous researchers have explored structure-based alignment with success to certain extent. The authors of Carnelian chose the reverse direction, which is interesting (though not thoroughly explored in this article).

>> We agree with the reviewer that including structural information in protein function prediction is indeed valuable. Unfortunately, we currently do not have much structural information available for prokaryotic proteins. Once we have more and more structurally resolved prokaryotic proteins, this will definitely be an interesting direction to pursue. We acknowledge this point in the Background section of the updated manuscript.

However, I still have multiple concerns of the revised work and the responses to my questions—some of which are substantial—as detailed below.

Novelty. I am not convinced that Carnelian represents a substantial innovation over current methodology. Cross-sample comparison has been done by many, using for example HUMAnN2 and other tools. Strong biological signals have been demonstrated in previous publications. The Carnelian team did a good job in demonstrating the advantage of their program in this task on selected datasets. Yet, I see this advantage as incremental, not game-changing. Second, I don't think the exploitation of k-mer matching in protein alignment is unprecedented. The authors already noted Kraken2. And I want to point the authors to a recently published protein sequence searcher: MMSeqs2 (Steinegger & Söding, 2017, Nat Biotech), which is gaining popularity in the field. The program uses a multiple layer engine, with the first of which being similar-k-mer matching (though not gapped), followed by ungapped and gapped sequence alignment.

>> While cross-sample comparison has been done by many, Carnelian's uniqueness and impact comes from finding concordant results in *cross-population* comparisons of healthy and disease individuals, including cohorts of disease samples from different populations and healthy vs. disease populations. While k-mer matching has been exploited in protein alignment, as the reviewer pointed out, this has been done only for exact k-mer matching. Carnelian's gapped k-mer based feature representation is new to the field, and has the advantage of not needing an explicit alignment in order to predict functionality. We now provide comparisons with MMSeqs2, PHMMER, and DIAMOND to show the advantage of Carnelian's gapped k-mer based classification over alignment-based protein search tools. These are newly included in the Supplemental Benchmarking Experiments (Additional File 1). Additionally, we now mention the advantage of not needing explicit alignment in the Background section.

Comparison with other tools. The authors made efforts to justify the choice of HUMAnN2 (native and uniform databases), Kraken2 and mi-faster to benchmark against Carnelian. They make some sense and I appreciate that. However I am not convinced that the current strategy alone is most meaningful. These tools being complex functional profilers (with the exception of Kraken2) naturally fail to satisfy the necessity of variable controlling in a scientific experiment design. As the authors admitted, this is not an "apple-to-apple" comparison. I see the marketing value but I am afraid that the scientific rigor is impaired.

>> We want to emphasize that, the nature of the task itself is such that it is difficult to do an "apple-to-apple" comparison. We tried our best to ensure the scientific rigor of our comparisons. In terms of variable controlling in a scientific experiment design, we created a uniform gold standard database to compare against and used the default parameters of each method. For the downstream differential abundance analyses the same normalization technique was applied to the raw counts profiles generated by the tools and the same pathway database was used with all the tools. We are unsure of what else we could have done.

In my opinion, the most relevant tools to include in benchmarking are the gapped k-mer binner (backend of Carnelian), DIAMOND (sequence alignment), Kraken2 (k-mer matching), HMMER (profile HMM), and MMseqs2 (combined search engine). One shouldn't simply assume "DIAMOND is the state-of-the-art" and skip this basic step.

With this experiment completed, I think the comparison of complex functional profilers is more reasonable, as it informs the users what they can get from each tool out-of-the-box. I appreciate the authors' addition of HUMAnN2 out-of-the-box mode in the comparison. I am also curious: what about PICRUSt (Langille et al., 2013), which predicts function simply based on taxonomy? We are aware of its limitation in accuracy due to say horizontal gene transfer (as the Carnelian authors also mentioned). But regarding the cross-sample comparison task, I am curious how it will perform.

>> We now provide comparisons with MMSeqs2, PHMMER, and DIAMOND to show the advantage of Carnelian's gapped k-mer based classification over protein aligners which are designed to detect sequence homologs. These are newly included in the Supplemental Benchmarking Experiments (Additional File 1). Please note that these tools are not designed for either functional annotation of protein sequences or functional profiling of shotgun read datasets. They are mainly used for searching sequence homologs of a query protein in a target protein database. Comparison with Kraken2 is already provided in our manuscript. We hope these comparisons are sufficient. PICRUSt is designed to work with 16S rRNA sequences or marker genes. Hence, comparison with PICRUSt is not appropriate here.

Sensitivity and spurious labels: The authors addressed my concern with de-emphasization of sensitivity (though left some, for example page 4 line 34) and discussions on the issue of spurious labels. I appreciate this but my concern remains with the revised form. The 2nd last paragraph of page 9 have multiple “we believe that...” I think these are over-claims that are misleading, especially when the lower precision of Carnelian is already evident.

>> We have toned down our claims regarding sensitivity in the revised Discussion section. However, although we no longer emphasize sensitivity as a chief selling feature, it is still a characteristic. We left the mentions in Results (e.g. page 4 line 34) because it is a salient feature, which we then discuss in Discussion with the caveat that sensitivity might just be spurious labels.

More problematic is page 3 line 21: “classify patients vs control ... Carnelian classifications are not spurious”. I do not see the logic here. I understand that experimental validation is difficult, but it is necessary if one wants claim against spurious labels. Therefore I would suggest the authors focus on cross-sample comparability but don't over claim accuracy.

>> We have rephrased the sentence in the revised manuscript to make our intent clear. In our patient-vs-control classification experiments, we used differentially abundant ECs detected by Carnelian, as well as other methods as features. In a classification experiment, if a method achieves higher accuracy, it implies the features learned from the data are meaningful and can differentiate better between the two classes. Since we achieve significantly higher accuracy (AUC) using the differentially abundant ECs detected by Carnelian compared to other methods, one can naturally conclude that the ECs we detect are more meaningful in differentiating between patients and controls. If they were just spurious hits, one wouldn't expect to achieve such high accuracy. We now provide this explanation in Results, where we make the claims.

Stratification: The authors presented intense reasoning for not performing taxonomic binning of functional units in Carnelian. I partially agree. What I agree with is that stratifying functional units by taxonomy is a difficult problem, and HUMAnN2's solution is far from being perfect. HUMAnN2 has its inherited limitation by adopting MetaPhlAn2, which is marker gene based, with high precision but low sensitivity comparing to alternatives (Ye et al., 2019, Cell). I also agree that stratification can be assessed by pairing Carnelian with a taxonomic profiler.

Though, admittedly, Carnelian's without taxonomic stratification (while HUMAnN2 has) is a limitation for users, many of whom do have this demand.

>> Carnelian's goal is to provide a framework for functional profiling that makes downstream comparative studies meaningful. While there is a demand for taxonomic profiling in the field, we believe it is not always the best idea to provide an all-encompassing solution that attempts to solve every problem of interest. That said, we have provided taxonomic profiling in Luo, Yu, et al. 2018 (OPAL), which uses a gapped k-mer classifier for taxonomic binning (same as the “backend” of Carnelian); thus, users can use OPAL for that task. We make this point clear in Conclusion.

The authors were right that a large number of studies have focused on aggregate functional profiles. But another large number of studies involving shotgun metagenomics have explored genome-level variation, using read-based or assembly-based techniques, the latter category of which has increasingly becoming practicable and standardized. I would thank the authors for pointing me to Pasolli et al., 2019, Cell. However, I am afraid that their interpretation of this important work is incorrect. Pasolli et al.'s starting materials were species-level genome bins (SGBs), that is, they were already assembled and binned (a process the Carnelian authors

deemed difficult), thus stratification was not necessary. Actually, Pasolli et al. Intensively discussed the characteristics and variation at genome level, including functional profile, as evident in their Tables S6 and S7.

>> We agree with the reviewer that some recent studies tried to look at stratified functional profiles using assembly-based techniques. Those are interesting when one is performing an exploratory study of a new metagenome. However, in comparative metagenomics for health and disease, the usefulness of stratified profiles is still limited. The Pasolli et al. study attempts to characterize a large number of previously unexplored genomes where it made sense. However, if we take all the limitations of metagenomic assembly into account, there is room for reasonable doubt whether the SGBs they find are meaningful, especially in the context of disease pathology. Also tables S6 and S7 in the Pasolli et al. paper (mentioned by the reviewer) indicate only the presence or absence of KEGG terms in SGBs, not relative abundances. Treating those binary vectors as functional profiles is simplistic and comparing them across populations might not be as meaningful as comparing aggregate profiles with relative abundances.

Effective protein length: I appreciate the authors' additional work in elucidating and validating the effectiveness of this metric. The explanation is very clear and helpful. This notion is overall fine to me. I agree that comparing to more complicated math, the simplicity and interpretability of effective protein length is an attraction, especially in the field of functional profiling where advanced maths have not been thoroughly explored.

However, I hope the authors note this question in the manuscript. It is a crucially important, yet not well-answered question. Carnelian's treatment, which considers per-sample relative abundance of functional proteins, is at the edge of the compositionality trap, and could also be confusing to some people (why functional units add up to one?). I hope the users are informed, and more ideally, have the option of keeping intermediate raw hit counts, on which users can experiment various maths.

>> We agree with the reviewer that the users should be informed and have the option of keeping intermediate raw hit counts. Indeed, this is the reason that Carnelian already provides both raw counts and normalized counts as outputs (see online user manual; github readm.txt line 111). In this way, a more sophisticated user can experiment with various normalizations. We've now added a note about it on the Carnelian website as well.

However, we make the normalized counts the default, because as the reviewer likely realizes, the reason to make per sample relative abundances sum upto 1 million is to put the abundances on the same scale so that they are directly comparable. Say, relative abundance of EC X in Samples A and B are 0.15 and 0.30, respectively. These quantities can be directly compared only if total relative abundance of all ECs in Samples A and B are the same. Otherwise, we do not really have a way of telling whether EC X is more abundant in A or B. We now clarify this point in Methods where we introduced the normalization step.

Synthetic metagenome. The protocol for designing the human gut microbiome sounds overall fine to me. But the authors missed two things: 1) "top 20 most abundant species" does not mean a typical gut microbiome is mainly composed of these species together. 2) Sorting species name by alphabetical order and applying a simple geometric distribution may not be ideal.

The species list in Table TX6 are indeed common human gut associates. What I questioned was the distribution of relative abundances. Alistipes can't be this high (60+%). Prevotella, Ruminococcus and Faecalibacterium are typically higher than 0.0x%. There should be several percentages of Actinobacteria and Proteobacteria. Actually, the last of which is among the main

sources of profiling challenge due to their genomic flexibility and pathogenic potentials. (There are also several dozen percentages of dark matters, but let's not complicate things here.) I would suggest the authors consult Eric Alm and his crew again with this specific question.

>> Notably, the relative abundances for the top 20 most abundant species were selected as described in the HUMAnN2 paper. In light of the reviewer's suggestion, we have now consulted Dr. Alm and colleagues on this question and newly added an experiment on another synthetic metagenome which is composed of 63% Bacteroidetes (~55% Bacteroides and 8% Alistipes), 6.4% Faecalibacterium, 5.4% Eubacterium, 12.2% Clostridiales, 5% Proteobacteria, 6% Actinobacteria, 1% Fusobacteria, and 1% Verrucomicrobia. This composition closely follows the average proportions of the observed gut microbial genera reported by Louis et al., 2016. The favorable performance of Carnelian vs other methods on this new metagenome is now provided in the Supplementary Benchmarking Experiments (Additional File 1).

Multiple assignments and reliance on EC terms. I accept the authors' reasoning, considering that Carnelian mainly predicts enzymes, and that multiple existing tools and protocols also don't resolve this issue well, for example they only take the hit with best alignment score, or simply the first hit come across in database search. I also appreciate that the authors generated and provided the COG model.

>> We thank the reviewer for appreciating our reliance on EC's and accepting our reasoning.

PERMANOVA. I do not agree with the authors' defense against reporting R2 value in a PERMANOVA test. I understand that some people don't like R2, just as some people (likely many more) don't like p-value (for example see Amrhein et al., 2019, Nature: Scientists rise up against statistical significance). Nevertheless, I don't know any decent journal would not demand reporting p-value when available. Admittedly, R2 is by far the most widely used and basic metric for assessing the effect size of variables. Therefore, I would still recommend that the authors report R2 values, together with pseudo-F and p-values.

>> Thanks for the suggestion. With respect to R2 values, we now report them along with the p-values and pseudo-F values in the updated Figure 3.

Unpublished datasets. My understanding is that the Bostonian and Baka datasets have not been published, and the authors have no intention to make them publicly accessible. This blocks the chance for readers to replicate the bioinformatics described in this article. I am not sure if this practice complies with the Genome Biology policy. But I will leave this question to the editor.

>> The Bostonian data is now published (Poyet et al., Nature Medicine, 2019) and the study and sample accession numbers are provided in the updated manuscript (Additional File 7). The data from Baka population is sensitive because Baka people are indigenous. Hence, these data are required to be released only through an access controlled repository; we cannot make these data available publicly. The Broad Institute is currently preparing them to be released through dbGaP; we will work out a solution with the editor about what to do in the interim.

Evaluation. Has the author satisfactorily responded to your previous review?

Reviewer #1: Yes

Reviewer #2: Yes